# Fixation can change the appearance of phase separation in living cells

Shawn Irgen-Gioro[1†], Shawn Yoshida[1,2†], Victoria Walling[1], Shasha Chong[1]*

[1]Division of Chemistry and Chemical Engineering, California Institute of Technology, Pasadena, United States; [2]Division of Biology and Biological Engineering, California Institute of Technology, Pasadena, United States

**Abstract** Fixing cells with paraformaldehyde (PFA) is an essential step in numerous biological techniques as it is thought to preserve a snapshot of biomolecular transactions in living cells. Fixed-cell imaging techniques such as immunofluorescence have been widely used to detect liquid–liquid phase separation (LLPS) in vivo. Here, we compared images, before and after fixation, of cells expressing intrinsically disordered proteins that are able to undergo LLPS. Surprisingly, we found that PFA fixation can both enhance and diminish putative LLPS behaviors. For specific proteins, fixation can even cause their droplet-like puncta to artificially appear in cells that do not have any detectable puncta in the live condition. Fixing cells in the presence of glycine, a molecule that modulates fixation rates, can reverse the fixation effect from enhancing to diminishing LLPS appearance. We further established a kinetic model of fixation in the context of dynamic protein–protein interactions. Simulations based on the model suggest that protein localization in fixed cells depends on an intricate balance of protein–protein interaction dynamics, the overall rate of fixation, and notably, the difference between fixation rates of different proteins. Consistent with simulations, live-cell single-molecule imaging experiments showed that a fast overall rate of fixation relative to protein–protein interaction dynamics can minimize fixation artifacts. Our work reveals that PFA fixation changes the appearance of LLPS from living cells, presents a caveat in studying LLPS using fixation-based methods, and suggests a mechanism underlying the fixation artifact.

*For correspondence:
schong@caltech.edu

†These authors contributed equally to this work

Competing interest: The authors declare that no competing interests exist.

## Editor's evaluation

Chemically fixing cells for fluorescence microscopy is a common practice in cell biology. However, fixation artifacts can lead the incorrect interpretations of experimental results. This article presents compelling evidence showing that in the context of liquid condensates formed by liquid–liquid phase separation (LLPS), paraformaldehyde (PFA) fixation creates a number of artifacts – such as changes in the number, appearance, or disappearance of liquid condensates. These important findings will be of great interest not only for those in the LLPS field but for any cell biologists using fixed samples for microscopy.

## Introduction

Fixing cells to preserve a snapshot of biomolecular transactions in vivo is a widely used strategy in numerous techniques in biology and medicine. Due to its small size and high reactivity with a wide range of biological entities, paraformaldehyde (PFA) is one of the most commonly used fixatives to create covalent cross-linking between biomolecules, for example, proteins and nucleic acids. PFA nonselectively 'fixes' or cross-links molecules in proximity to enable characterization of biomolecular interactions formed in living cells. Examples of popular techniques that use PFA to fix cells include ChIP-sequencing (*Robertson et al., 2007*; *Solomon and Varshavsky, 1985*), chromosome

**eLife digest** A typical human cell is a crowded soup of thousands of different proteins. One way that the cell organizes this complex mix of contents is by creating separate droplets within the cell, like oil in water. These droplets can form through a process known as liquid-liquid phase separation, or LLPS, where specific proteins gather in high concentrations to carry out their cellular roles.

The critical role of LLPS in cellular organization means that it is widely studied by biologists. To detect LLPS, researchers often subject the cells to treatments designed to hold all the proteins in place, creating a snapshot of their natural state. This process, known as fixing, allows scientists to easily label a protein with a fluorescent tag, take pictures of the cells, and look at whether the protein forms droplets in its natural state. This is often easier to do than imaging cells live, but it relies on LLPS being well-preserved upon fixation.

To test if this is true, Irgen-Gioro, Yoshida et al. looked at protein droplets in live cells, and then fixed the cells to check whether the appearance of the droplets had changed. The images taken showed that fixation could alter the size and number of droplets depending on the protein being studied. To explain why the effects of fixing change depending on the protein, Irgen-Gioro, Yoshida et al. hypothesized that a faster fixation – relative to how quickly proteins can bind and unbind to their droplets – can better preserve the LLPS droplets. They verified their idea using a microscopy technique in which they imaged single molecules, allowing them to see how different fixation speeds relative to protein binding affected the droplets.

The work of Irgen-Gioro, Yoshida et al. identifies an important caveat to using fixation for the study of LLPS in cells. Their findings suggest that researchers should be cautious when interpreting the results of such studies. Given that LLPS in cells is an area of research with a lot of interest, these results could benefit a broad range of biological and medical fields. In the future, Irgen-Gioro, Yoshida et al.'s findings could prompt scientists to develop new fixing methods that better preserve LLPS in cells.

conformation capture (3C)-based techniques (*Dekker et al., 2002*), immunofluorescence (*Richter et al., 2018*), fluorescence in situ hybridization (FISH) (*Moter and Göbel, 2000*), cross-linking mass spectrometry (*Sutherland et al., 2008*), super-resolution expansion microscopy (*Chen et al., 2015*), and super-resolution localization microscopies such as stochastic optical reconstruction microscopy (STORM) (*Rust et al., 2006*). Although PFA fixation has been used to faithfully preserve live-cell conditions in many scenarios, a number of studies have uncovered situations in which fixation fails to cross-link DNA–protein interactions formed in living cells. By imaging different transcription factors (TFs) in live and fixed cells, *Schmiedeberg et al., 2009* showed that TFs bound to DNA with fast dissociation dynamics (<5 s residence times as determined by fluorescence recovery after photobleaching [FRAP]) are not cross-linked to DNA upon PFA fixation. Using live-cell single-molecule imaging, *Teves et al., 2016* showed that TFs stay bound to chromosome during mitosis and fixing cells can artificially deplete transiently bound TFs from mitotic chromosomes. These studies exemplify the fact that fixation, with limited reaction rates, cannot provide an instantaneous snapshot and may miss or obfuscate biomolecular interactions that happen either at or faster than the timescale of fixation. What further complicates the result of cell fixation is that the reactivity and reaction rates of PFA are variable and dependent on its biomolecule substrates (*Gavrilov et al., 2015*; *Shishodia et al., 2018*). For example, the efficiency and rates at which PFA reacts with proteins can vary by orders of magnitude (*Kamps et al., 2019*) and are dependent on their amino acid sequences (*Kamps et al., 2019*; *Metz et al., 2004*; *Sutherland et al., 2008*) and tertiary structures (*Hoffman et al., 2015*).

Among the numerous biomolecular transactions investigated using fixed-cell imaging is liquid–liquid phase separation (LLPS), a long-observed behavior of polymers in solution (*Gibbs, 1879*; *Graham, 1861*; *Hyman et al., 2014*) that has recently generated much excitement in biological research communities due to its proposed roles in cellular organization and functions (*Banani et al., 2017*; *Boeynaems et al., 2018*; *Mitrea and Kriwacki, 2016*; *Shin and Brangwynne, 2017*). LLPS is driven by excessive levels of transient, selective, and multivalent protein–protein interactions mediated by intrinsically disordered regions (IDRs) within the proteins of interest (*Chong et al., 2018*; *Kato and McKnight, 2018*; *Li et al., 2012*). Whereas rigorous characterization of LLPS in vivo has been challenging and remains a question under active investigation (*McSwiggen et al., 2019b*),

detection of discrete puncta that have a spherical shape, undergo fusion and fission, and dynamically exchange biomolecules with the surrounding according to FRAP is often considered evidence of putative LLPS in living cells. While such diverse measurements have been widely used for studying proteins under overexpression conditions, far fewer approaches are available to probe LLPS under physiological conditions. Detecting local high-concentration regions or puncta of an endogenously expressed protein using immunofluorescence of fixed cells has been used in many studies as evidence of LLPS (*Boija et al., 2018*; *Guo et al., 2019*; *Owen et al., 2021*; *Xie et al., 2022*; *Yang et al., 2020*). Not only is the detection of puncta an inconclusive metric for establishing LLPS, whether a punctate distribution observed in fixed cells actually represents the live-cell scenario remains unclear as fixation has only been assumed, but not directly shown, to faithfully preserve multivalent interactions and LLPS formed in living cells. This knowledge gap motivated us to image cells that overexpress various known IDR-containing proteins before and after fixation to evaluate the ability of PFA fixation to preserve LLPS behaviors. We found that, interestingly, fixation can significantly alter the appearance of droplet-like puncta in cells. Our quantitative image analysis suggests that depending on the LLPS-driving protein, fixing cells can either enhance or diminish the apparent LLPS behaviors in vivo. In certain cases, fixation can even cause droplet-like puncta to artificially appear in cells that have a homogeneous protein distribution and no detectable puncta in the live condition. Conversely, fixation can also cause droplet-like puncta in living cells to completely disappear. Combining experiments that modulate fixation rates, live-cell single-molecule imaging that quantifies protein binding dynamics, and simulations based on a kinetic model, we further demonstrated that protein localization in fixed cells depends on an intricate balance of protein–protein interaction dynamics, the overall rate of fixation, and the difference between protein fixation rates in and out of droplet-like puncta. Our work urges caution in the interpretation of previous claims of in vivo phase separation based solely on immunofluorescence imaging of fixed cells and serves to guide future judicious application of PFA fixation.

## Results

### Fixation enhances the LLPS appearance of FET family proteins

To investigate the effect of PFA fixation on the appearance of LLPS, we first compared confocal fluorescence images of live and fixed U2OS cells that transiently express an IDR tagged with EGFP and a nuclear localization sequence (NLS). We focused on the FET family protein IDRs (AA2-214 of FUS, AA47-266 of EWS, and AA2-205 of TAF15) that are reported to undergo putative LLPS in cells upon overexpression (*Altmeyer et al., 2015*; *Chong et al., 2018*; *Wang et al., 2018*). *Figure 1*, *Video 1*, and *Figure 1—figure supplement 1* compare the same cells before and after treatment of 4% PFA for 10 min unless otherwise noted, a typical condition utilized for fixed-cell imaging techniques such as immunofluorescence. At high enough expression levels, all three IDRs are able to form discrete and spherical puncta in the live cell nucleus, which show fusion and fission behaviors and are thereby consistent with LLPS droplets (*Alberti et al., 2019*; *Banani et al., 2017*). Interestingly, after fixation, the puncta of all three IDRs appear to increase in their numbers, sizes, and contrast compared with the dilute phase. In particular, PFA fixation was able to artificially turn a cell with EGFP-EWS(IDR) homogeneously distributed in the nucleus without any puncta into one with many discrete puncta (*Figure 1*). We quantified the fixation-induced changes of LLPS appearance by calculating three parameters from the fluorescence images of cells, including the number of puncta, surface roughness, and punctate percentage, and found a significant increase in all three parameters after fixation (*Figure 1D–F*, *Figure 1—source data 1*). The number of puncta and punctate percentage (percentage of intra-nuclear fluorescence intensity in the concentrated phase) are indicators of the propensity to phase separate (*Berry et al., 2015*). The surface roughness (standard deviation of pixel intensities across the nucleus) quantifies the uneven distribution of a fluorescently labeled protein in the nucleus, allowing for detection of puncta appearance or disappearance without the need for an algorithm to identify individual puncta in the cell.

We next tested how the fixation artifact is dependent on the length of PFA treatment, PFA concentration, and the type of fixatives. We performed real-time imaging of live cells expressing EGFP-FUS(IDR) and found that their morphology and LLPS appearance start to change immediately upon PFA treatment and reach a steady state after ~100 s of treatment (*Video 1*, *Figure 1—figure supplement 2*). We treated cells expressing EGFP-EWS(IDR) with different concentrations of PFA (1, 2, 4,

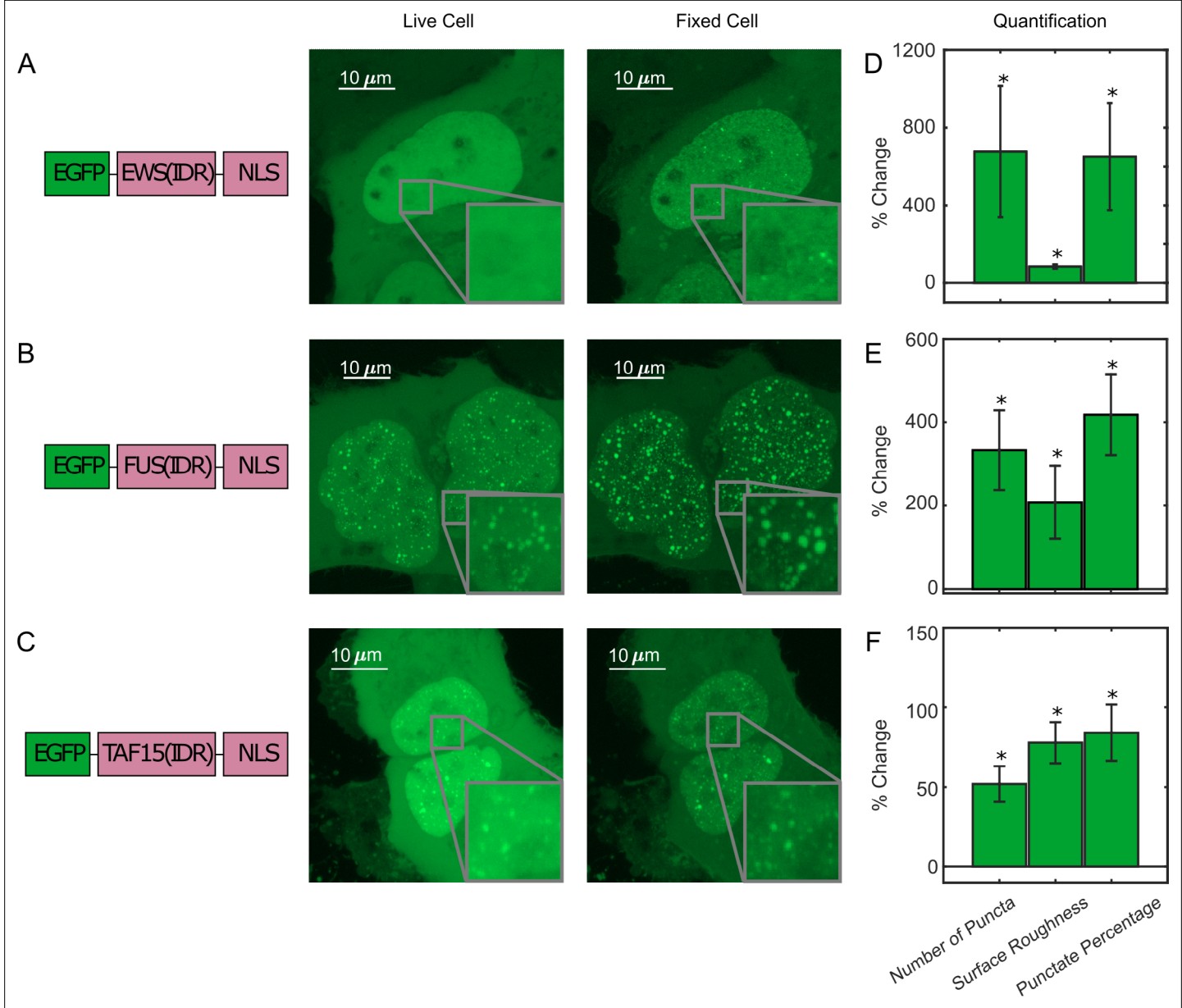

**Figure 1.** Fixation can change the apparent liquid–liquid phase separation (LLPS) behaviors of proteins. (**A**) EGFP-EWS(IDR), (**B**) EGFP-FUS(IDR), and (**C**) EGFP-TAF15(IDR) are transiently expressed in U2OS cells and imaged before and after fixation using confocal fluorescence microscopy. A schematic of each protein construct is shown on the left. A maximum z-projection of a representative live cell expressing its respective protein is shown next to that of the same cell after 10 min of fixation with 4% paraformaldehyde (PFA). The inserts show a zoomed-in region of the cell. (**D–F**) Quantification of percentage change of LLPS parameters after fixation. The values are averaged from 34 (**D**), 17 (**E**), or 24 (**F**) cells measured in 3 (**D**), 2 (**E**), or 2 (**F**) independent transfection and imaging sessions. Error bars represent standard errors. Asterisks indicate a significant difference compared with 0 ($p < 0.05$, Wilcoxon signed-rank test).

The online version of this article includes the following source data and figure supplement(s) for figure 1:

**Source data 1.** Quantification of puncta parameters used to generate the bar plots.

**Figure supplement 1.** EGFP-EWS(IDR) can form droplet-like puncta in living cells, which change appearance upon fixation.

**Figure supplement 2.** Quantification of *Video 1* shows the number of EGFP-FUS(IDR) puncta in the cell as a function of the length of paraformaldehyde (PFA) treatment.

**Figure supplement 3.** Fixation at various paraformaldehyde (PFA) concentrations can change the apparent liquid–liquid phase separation (LLPS) behaviors of EGFP-EWS(IDR).

**Figure supplement 3—source data 1.** Quantification of puncta parameters used to generate *Figure 1—figure supplements 3 and 4*.

*Figure 1 continued on next page*

*Figure 1 continued*

**Figure supplement 4.** Fixation using paraformaldehyde/glutaraldehyde (PFA/GA) in combination still changes the apparent liquid–liquid phase separation (LLPS) behaviors of EGFP-EWS(IDR).

and 8%) and observed statistically significant changes to the above three LLPS-describing parameters upon fixation at all the concentrations (*Figure 1—figure supplement 3*). PFA in combination with glutaraldehyde (GA) has been shown to reduce fixation artifacts in imaging the distribution of cell membrane receptors (*Stanly et al., 2016*). However, we still observed statistically significant fixation-induced changes to the apparent LLPS behavior of EGFP-EWS(IDR) using 4% PFA and 0.2% GA in combination (*Figure 1—figure supplement 4*).

We next compared the intracellular distribution of TAF15(IDR) tagged with different fluorescent tags, including, EGFP, DsRed2, and HaloTag, before and after fixation with 4% PFA. The LLPS behavior of DsRed2-TAF15(IDR) is enhanced upon fixation like EGFP-TAF15(IDR) (*Figure 2A*), but the enhancement has a different appearance. Whereas there is not a significant change to the large preformed DsRed2-TAF15(IDR) puncta, thousands of smaller puncta emerge in the dilute phase within the nucleus (*Figure 2B*). In contrast, Halo-TAF15(IDR) displays a diminished LLPS behavior after fixation, with its puncta becoming smaller and dimmer or completely disappearing (*Figure 2C*, *Figure 2—figure supplement 1*). Quantification of the number of puncta, surface roughness, and punctate percentage of the TAF15(IDR) LLPS systems before and after fixation further confirmed these observations (*Figure 2D–F*, *Figure 2—source data 1*). The fact that different phase-separating proteins can have bifurcating behaviors upon fixation is interesting. While it is known that EGFP and DsRed2 can dimerize and HaloTag cannot (*Costantini et al., 2012*; *Sacchetti et al., 2002*), it is unclear whether and how the dimerization potential might contribute to the proteins' bifurcating responses to PFA fixation. We note that the fixation-induced changes to LLPS appearance can affect the physical characterization of in vivo LLPS systems based on fixed-cell imaging, such as the Gibbs energy of transfer between dilute and concentrated phases (*Riback et al., 2020*) and how far from the critical concentration a system is (*Bracha et al., 2018*), potentially affecting the interpretation of the functional role of LLPS in cellular processes. Moreover, the fact that PFA fixation can artificially promote puncta formation even in cells without detectable puncta in the live condition presents an important caveat in fixation-based approaches that have been commonly used for characterizing LLPS under physiological conditions, for example, immunofluorescence.

Furthermore, to examine whether all phase-separating proteins show the fixation artifact, we compared live- and fixed-cell images of EGFP-tagged full-length FUS (FUS(FL)). Full-length FUS is reported to have a greater LLPS propensity in vitro than its IDR alone (*Wang et al., 2018*). We found that EGFP-FUS(FL) overexpressed in live U2OS cells forms many small puncta throughout the nucleus, and we did not observe a significant change of this behavior after PFA fixation (*Figure 3A*, *Figure 3—source data 1*). We also fused Halo-tagged TAF15(IDR) to FTH1 that forms a 24-mer (*Bellapadrona and Elbaum, 2014* and *Bracha et al., 2018*) to make an artificial protein with a high LLPS propensity. We found that TAF15(IDR)-Halo-FTH1 overexpressed in live U2OS cells forms large droplet-like puncta and the appearance of LLPS does not significantly change after PFA fixation (*Figure 3B*, *Figure 3—source data 1*). In addition, we looked into a native IDR-containing protein, EWS::FLI1, an oncogenic TF causing Ewing sarcoma (*Grünewald et al., 2018*) and known to form local high-concentration hubs at target genes associated with GGAA microsatellites (*Chong et al., 2018*). Although there is no convincing evidence that EWS::FLI1 undergoes LLPS under physiological conditions, the formation of its hubs is mediated by the homotypic multivalent interactions of EWS(IDR) within the protein. Excessive levels of such multivalent interactions often result in LLPS (*Li et al., 2012*). We previously Halo-tagged endogenous EWS::FLI1 in an Ewing sarcoma cell line A673 using CRISPR/Cas9-mediated genome editing (*Chong et al., 2018*). Here, we compared live and fixed A673

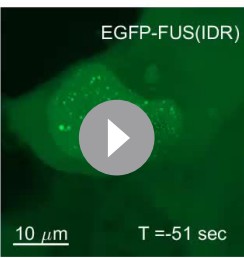

**Video 1.** Real-time imaging of a U2OS cell expressing EGFP-FUS(IDR) during paraformaldehyde (PFA) fixation. https://elifesciences.org/articles/79903/figures#video1

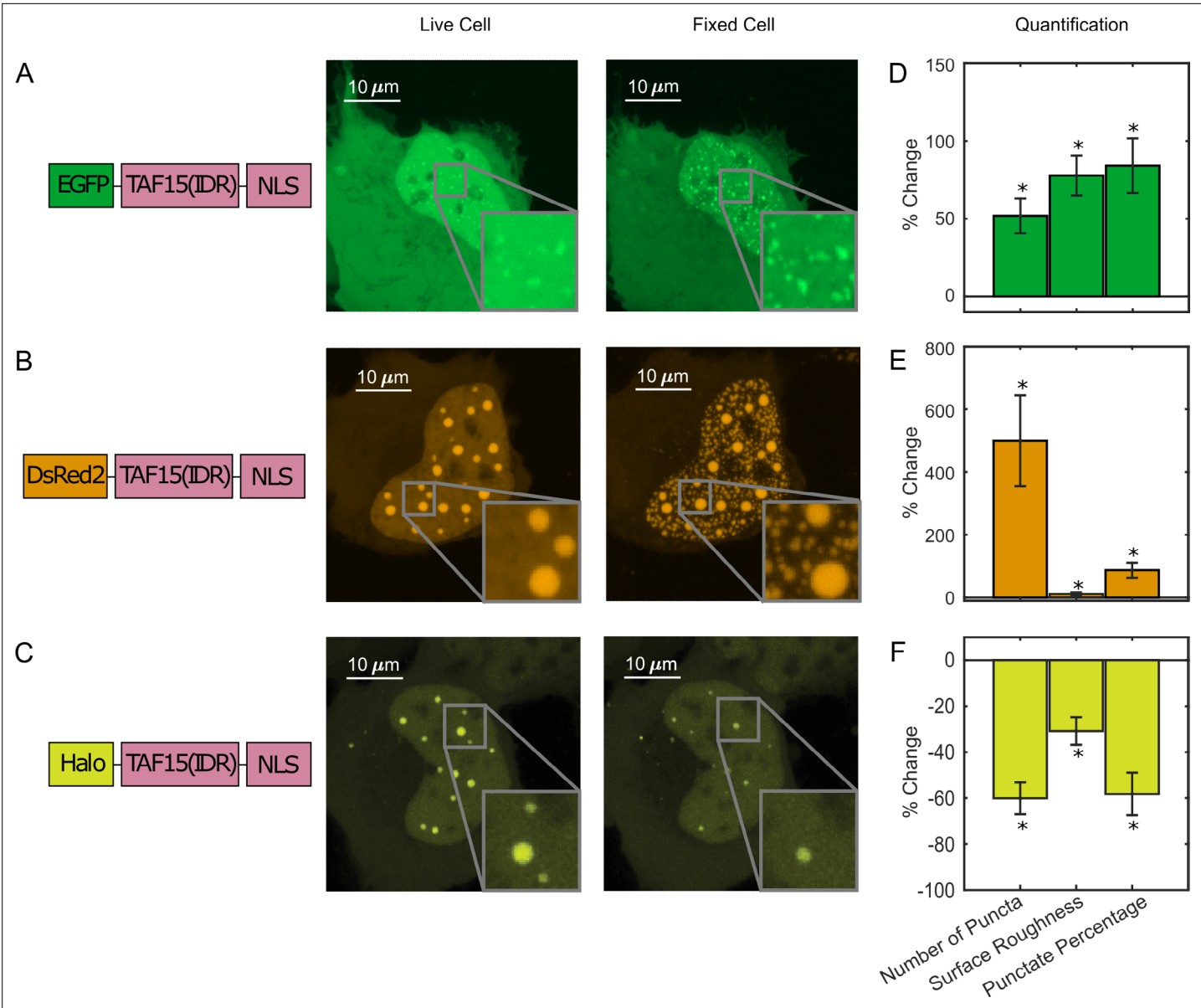

**Figure 2.** Paraformaldehyde (PFA) fixation can both enhance and diminish liquid–liquid phase separation (LLPS) appearance. U2OS cells expressing (**A**) EGFP-TAF15(IDR), (**B**) DsRed2-TAF15(IDR), and (**C**) Halo-TAF15(IDR), ligated with the JFX549 Halo ligand, are imaged using confocal fluorescence microscopy before and after 10 min of fixation with 4% PFA. Schematics of the protein constructs are shown on the left. Live- and fixed-cell images are compared. (**D–F**) Quantification of LLPS parameters after fixation. The values are averaged from 24 (**D**), 23 (**E**), or 10 (**F**) cells measured in 2 (**D**), 2 (**E**), or 3 (**F**) independent transfection and imaging sessions. Error bars represent standard errors. Asterisks indicate a significant difference compared with 0 ($p < 0.05$, Wilcoxon signed-rank test).

The online version of this article includes the following source data and figure supplement(s) for figure 2:

**Source data 1.** Quantification of puncta parameters used to generate the bar plots.

**Figure supplement 1.** Fixation can diminish liquid–liquid phase separation (LLPS) appearance.

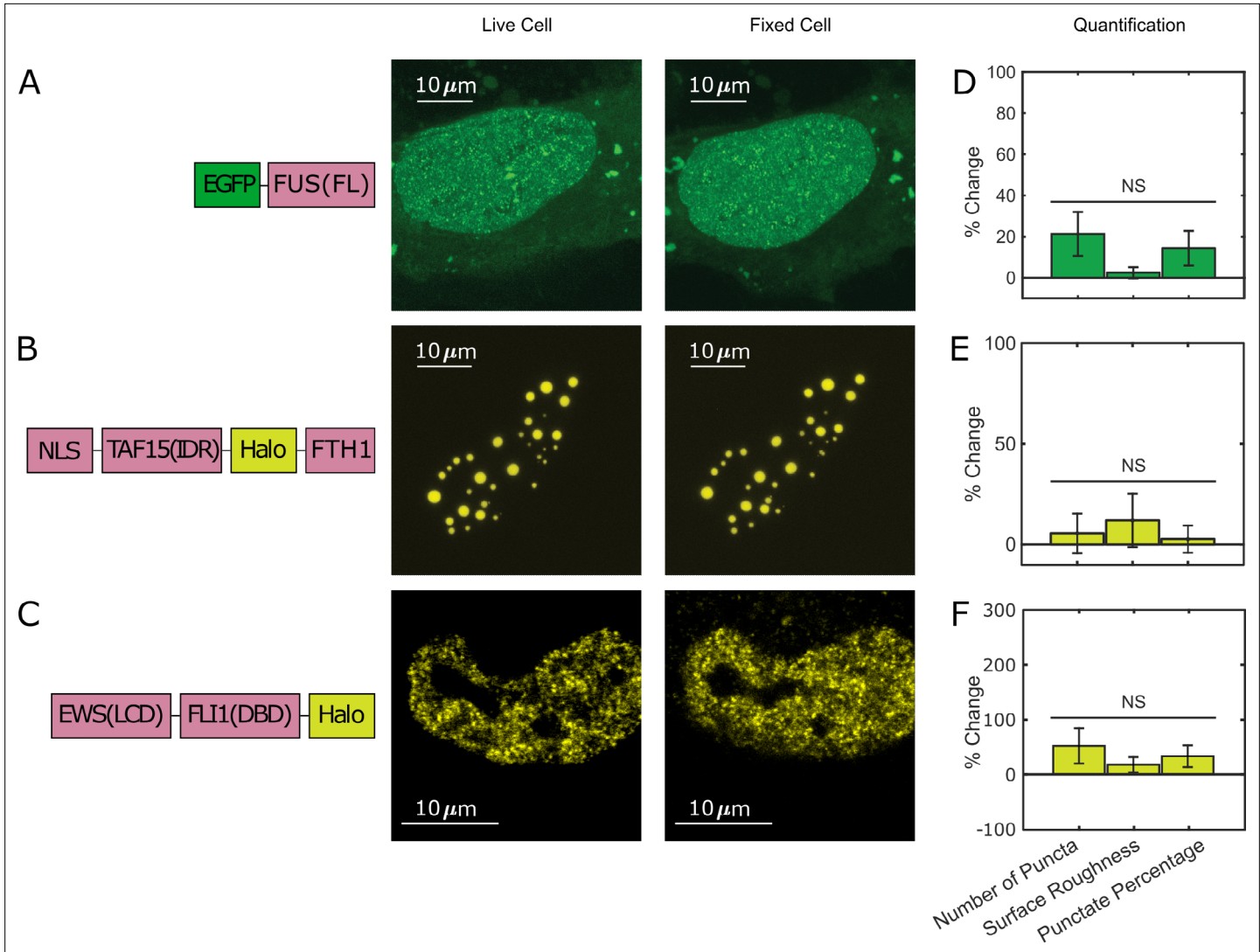

**Figure 3.** Not all puncta-forming proteins show the fixation artifact. U2OS cells expressing (**A**) EGFP-FUS(FL) and (**B**) TAF15(IDR)-Halo-FTH1, and (**C**) an A673 cell expressing endogenous EWS::FLI1-Halo are imaged using confocal fluorescence microscopy before and after 10 min of fixation with 4% paraformaldehyde (PFA). Halo-tagged proteins are ligated with the JFX549 Halo ligand before imaging. Schematics of the protein constructs are shown on the left. Live- and fixed-cell images are compared. (**D–F**) Quantification of puncta parameters after fixation. The values are averaged from 21 (**D**), 16 (**E**), or 15 (**F**) cells measured in 1 (**D**), 4 (**E**), or 2 (**F**) independent transfection and imaging sessions. Error bars represent standard errors. NS: not significant difference compared with 0 (p<0.05, Wilcoxon signed-rank test). None of the examined proteins show significant changes in their liquid–liquid phase separation (LLPS) or hub appearance in the fixed-cell image as compared to the live-cell image.

The online version of this article includes the following source data for figure 3:

**Source data 1.** Quantification of puncta parameters used to generate the bar plots.

cell images of endogenous EWS::FLI1-Halo and did not observe a significant difference in its distribution (*Figure 3C*, *Figure 3—source data 1*). This result suggests that PFA fixation does not change the intracellular distribution of all proteins that have a LLPS potential.

## Switching between enhancing and diminishing the LLPS appearance depends on fixation kinetics

To understand what factors are underlying the diverging fixation artifact of in vivo LLPS systems, we performed the above-described fixation imaging assay with glycine added to live cells prior to PFA fixation. Glycine is highly reactive with formaldehyde and is commonly used to quench the formation of protein–protein cross-linked complexes by quickly forming protein–glycine and glycine–glycine

cross-linked adducts instead (*Hoffman et al., 2015*). We thus utilized additional glycine to generate a competitive fixation reaction in the cell against protein–protein fixation. We found that adding 25 mM glycine to live U2OS cells that overexpress DsRed2-TAF15(IDR) increases the starting punctate percentage from 18 ± 1.92 to 36 ± 3.82% (quantified from 23 cells), indicating an increase in the degree of LLPS. Although the underlying mechanism of such increase is unclear, we speculate this might be because hydrophobic intermolecular contacts that play an important role in TAF15(IDR) LLPS (*Patel et al., 2017*) are enhanced by the presence of hydrophobic glycine. Importantly, addition of glycine dramatically reversed the fixation effect on the LLPS behavior of DsRed2-TAF15(IDR). Whereas PFA fixation in the absence of additional glycine enhances the LLPS appearance (*Figure 2B*, *Figure 4A*), in the presence of 25 mM glycine, fixation causes many of the smaller puncta formed in live cells to disappear completely and larger, preformed puncta to turn into a 'donut' shape, with the outline of the puncta still visible but the interior devoid of the protein (*Figure 4B*). None of these fixed-cell images are good representations of live cells, but it appears that glycine affects the critical parameters that control the divergent artifact of PFA fixation. The observation that the appearance of droplet-like puncta in fixed cells can be dramatically modified by the presence of glycine competition emphasizes that the kinetics of fixation can play an essential role in the appearance of LLPS in fixed cells.

## Kinetic modeling explains the fixation artifact

Given our observation that fixation kinetics are critical to the appearance of LLPS in fixed cells, we numerically simulated a four-state kinetic model (*Hoops et al., 2006*). As shown in *Figure 5A and B*, the model focuses on one protein of interest (POI), which before fixation can either be in state $S_1$ – 'in puncta' or $S_2$ – 'out of puncta.' Because POI molecules are dynamically exchanged in and out of puncta, the in-puncta percentage (punctate percentage) of POI is at an equilibrium determined by the ratio of the binding rate, $k_1$, and the dissociation rate, $k_2$ (*Pollard, 2010*). These are the average exchange rates between $S_1$ and $S_2$ and do not concern the potential spatial inhomogeneity in the rates at the molecular level. For example, individual POI molecules at the surface and interior of a punctum might dissociate with different rates, but our model does not differentiate these molecules. We define the moment that PFA is added as time zero ($t = 0$) and introduce two fixed states of POI, which are $S_3$ (POI cross-linked to proteins within puncta) with a fixation rate of $k_3$ and $S_4$ (POI cross-linked to proteins outside puncta) with a fixation rate of $k_4$. Because fixing to both $S_3$ and $S_4$ states are irreversible, when the cell is fully fixed long after addition of PFA ($t = \infty$), there is no longer any concentration in $S_1$ and $S_2$. The fixation artifact of an LLPS system can be represented as the absolute change in punctate percentage, or the ratio of in-puncta POI to total POI, after fixation:

$$
\begin{aligned}
\Delta Punctate\ Percentage\ &= Final\ Punctate\ Percentage - Initial\ Punctate\ Percentage \\
&= \left( \frac{[S_3]_{t=\infty}}{[S_3]_{t=\infty} + [S_4]_{t=\infty}} - \frac{[S_1]_{t=0}}{[S_1]_{t=0} + [S_2]_{t=0}} \right) * 100
\end{aligned} \tag{1}
$$

We hypothesized that the balance between interaction and fixation dynamics in a LLPS system causes the fixation artifact and tested the hypothesis by calculating Δ *Punctate Percentage* as a function of various kinetic and equilibrium parameters.

It is well-established that the dilute and concentrated phases of an LLPS system have different protein composition and concentrations (*Currie and Rosen, 2022*; *Koga et al., 2011*; *Nott et al., 2015*; *Yewdall et al., 2021*). The rate of fixation is known to vary with both factors by orders of magnitude, with the timescale of fixation ranging from seconds to hours (*Hoffman et al., 2015*; *Kamps et al., 2019*; *Metz et al., 2006*; *Metz et al., 2004*). Because protein–protein interactions that drive LLPS are highly dynamic with binding residence times in the range of seconds to tens of seconds (*Chong et al., 2018*), fixation likely happens with either lower or comparable rates than protein binding and dissociation. We thus first examined whether different fixation rates of POI in and out of puncta can cause a fixation artifact, assuming the overall fixation rates ($k_3 + k_4$) are slower than protein binding and dissociation, and how the fixation artifact may depend on intrinsic protein–protein interaction equilibrium. Specifically, we calculated Δ*Punctate Percentage* as a function of the starting punctate percentage and the relative in-puncta fixation rate ($k_3 : k_4$) when the relative overall fixation rate is constant (($k_3 + k_4$) : ($k_1 + k_2$) = 1:5) (*Figure 5C*). In the scenario where the rate of fixation is the same in and out of the puncta ($k_3 = k_4$), the live-cell equilibrium is perfectly preserved in fixed cells

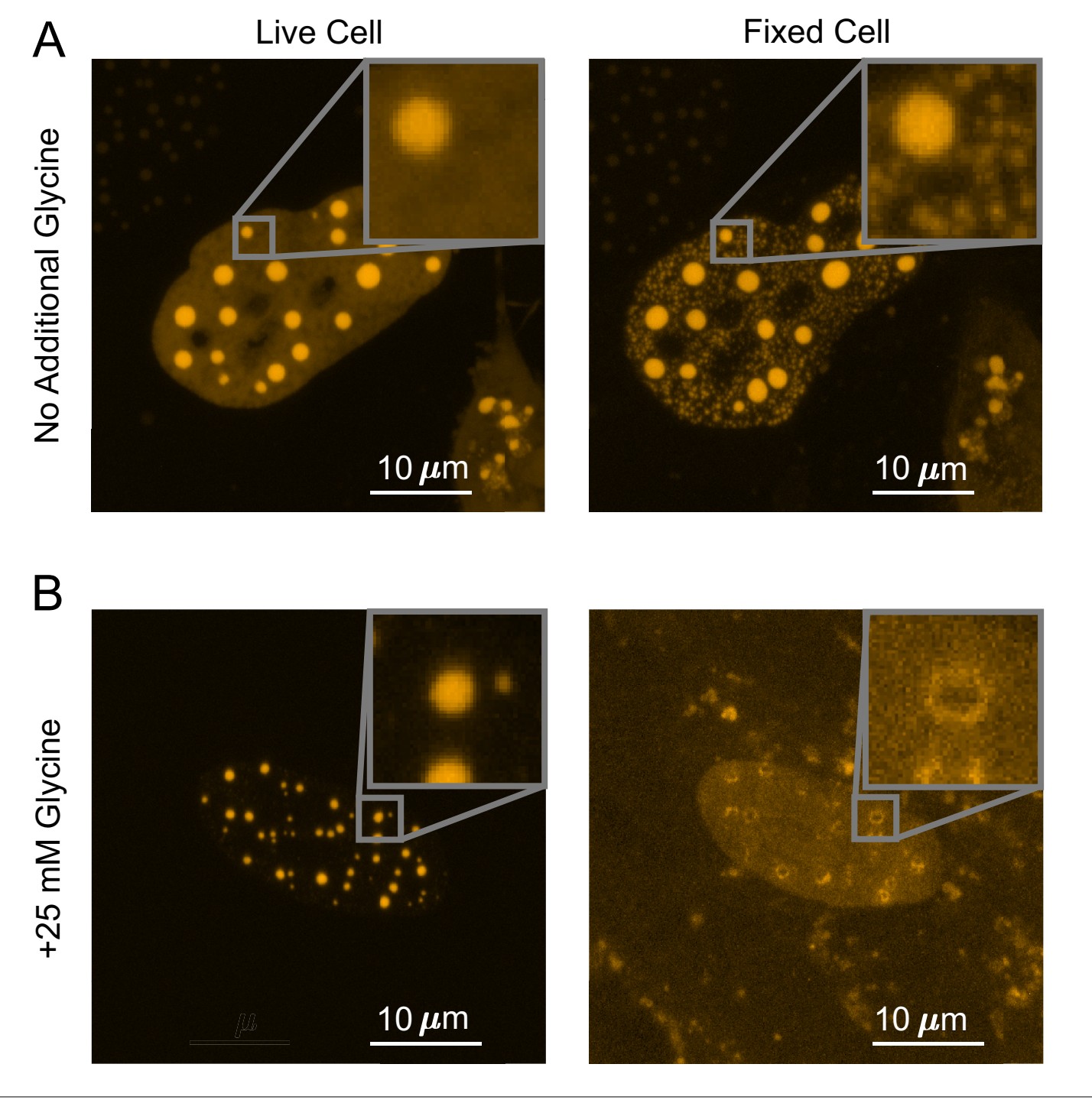

**Figure 4.** Competitive fixation pathway creates a reversed fixation artifact. (**A**) Fixing U2OS cells that express DsRed2-TAF15(IDR) in the absence of additional glycine causes many small puncta to appear. (**B**) Fixing cells in the presence of 25 mM additional glycine results in a reduction in the number of puncta, with large puncta forming 'donut' shapes. In both (**A**) and (**B**), cells are imaged using confocal fluorescence microscopy before and after 10 min of fixation with 4% paraformaldehyde (PFA).

regardless of the starting punctate percentage ($\Delta Punctate\ Percentage \sim 0$). However, when one fixation rate is faster than the other, we observe a bifurcating effect. When the fixation rate inside the puncta is greater than outside the puncta ($k_3 > k_4$), the fixed cell will have a higher punctate percentage than the live cell, that is, fixation enhances the apparent LLPS behaviors. When the balance is reversed

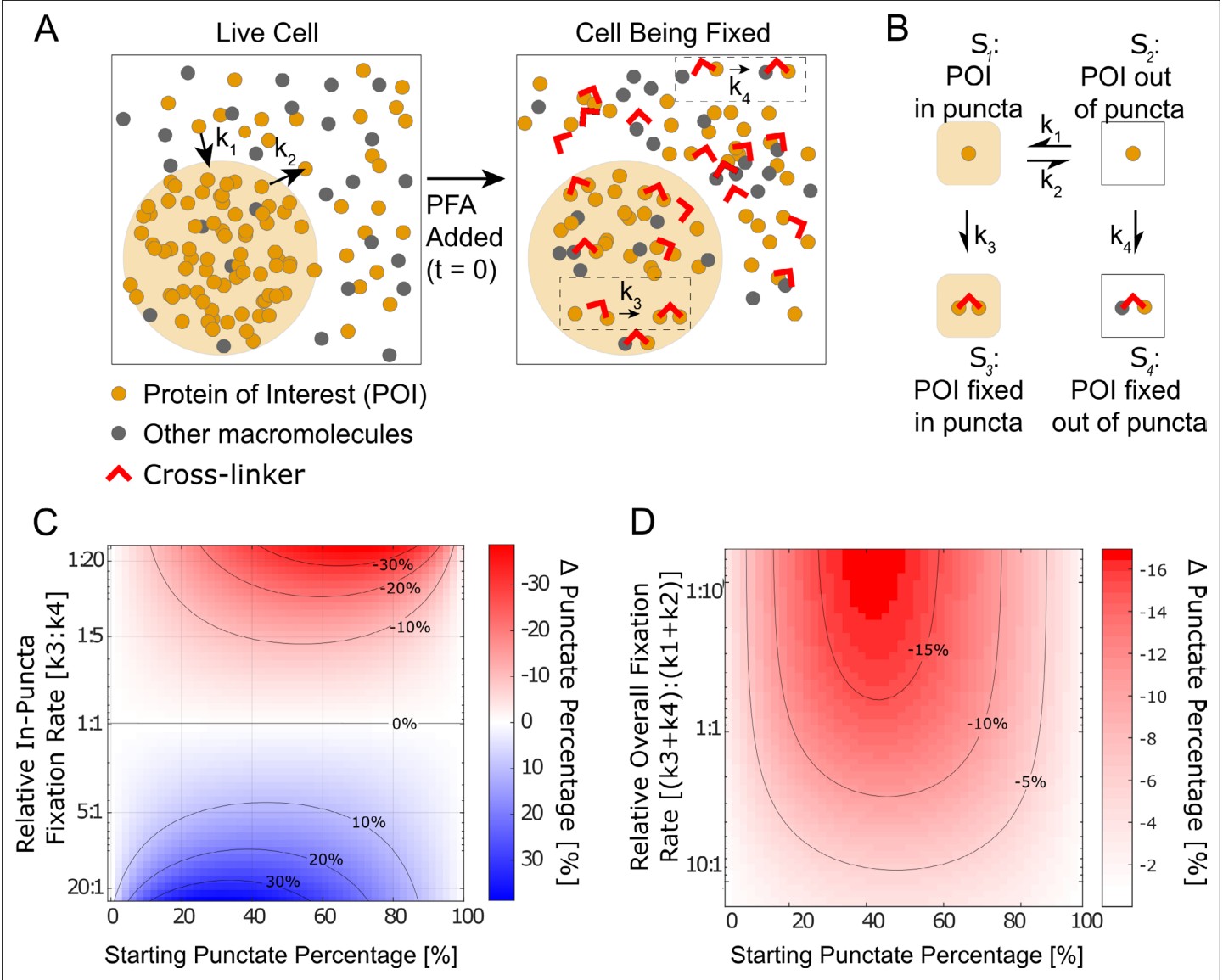

**Figure 5.** Kinetic simulation explains bifurcating fixation artifacts. (**A**) Schematic that describes fixation of a phase-separating protein of interest (POI) in the cell. (**B**) The four-state kinetic model with associated kinetic rates connecting the different states. (**C**) Simulation of the fixation artifact as a function of the starting punctate percentage and the relative in-puncta fixation rate $k_3 : k_4$, assuming the overall fixation rate as well as overall protein binding and dissociation rates are constant ($k_3 + k_4 = 0.2$, $k_1 + k_2 = 1$). Faster in-puncta fixation causes liquid–liquid phase separation (LLPS) behavior to be over-represented (blue). Slower in-puncta fixation causes LLPS behavior to be under-represented (red). (**D**) Simulation of the fixation artifact as a function of the starting punctate percentage and the relative overall fixation rate $(k_3 + k_4) : (k_1 + k_2)$, assuming individual fixation rates are constant ($k_3 = 1$, $k_4 = 2$). Fast overall fixation rate compared with protein–protein interaction dynamics decreases the fixation artifact. (**C**) and (**D**) were simulated over starting punctate percentages ranging from 0% ($k_1 = 0$, $k_2 = 1$) to 100% ($k_1 = 1$, $k_2 = 0$). Level curves are marked on (**C**) and (**D**).

($k_4 > k_3$), the fixed cells will have diminished apparent LLPS behaviors than in the live cell. For cases where the starting punctate percentage is near 0% or 100% due to significantly different POI binding and the dissociation rates ($k_2 \gg k_1$ or $k_1 \gg k_2$), no significant change to LLPS appearance happens after fixation ($\Delta Punctate\ Percentage \sim 0$). In short, our simulation suggests that having unequal fixation rates in and out of puncta is necessary to cause a fixation artifact of LLPS systems and the artifact is dependent on the punctate percentage of POI in living cells.

Because previous reports have documented that fixation preserves transient interactions worse than stable interactions (*Poorey et al., 2013*; *Schmiedeberg et al., 2009*; *Teves et al., 2016*), we next investigated how fixation rates relative to protein–protein interaction dynamics may impact the observed fixation artifact. Specifically, we calculated $\Delta Punctate\ Percentage$ as a function of both the

starting punctate percentage and the relative overall fixation rate, $(k_3 + k_4) : (k_1 + k_2)$, assuming a constant relative in-puncta fixation rate ($k_3 : k_4 = 1 : 2$) (*Figure 5D*). Here, a fast relative overall fixation rate can either be caused by slow protein–protein interaction dynamics (low $(k_1 + k_2)$) or fast absolute fixation rates (high $(k_3 + k_4)$). We found when the protein–protein interactions are highly dynamic compared with the overall fixation rates ($(k_3 + k_4) \ll (k_1 + k_2)$), the fixation artifact is the most pronounced as shown by a large value of $\Delta Punctate\ Percentage$. In contrast, when the protein–protein interactions are stable and less dynamic compared with the overall fixation rate ($(k_3 + k_4) \gg (k_1 + k_2)$), there is a minimal fixation artifact and the punctate percentage in fixed cells is similar to that in living cells ($\Delta Punctate\ Percentage \sim 0$). In short, our simulation suggests that when the overall fixation rate is fast compared with the dynamics of targeted interactions, fixation artifacts can be minimized even with unequal fixation rates in and out of puncta.

Overall, our kinetic model suggests that the observed fixation artifact of LLPS systems is driven by the interplay of three factors: protein–protein interaction dynamics, the absolute overall fixation rate, and different fixation rates in and out of puncta. Different fixation rates of POI in and out of puncta ($k_3 : k_4 \neq 1 : 1$) are required for fixation artifacts to happen and the value of $k_3 : k_4$ determines whether the LLPS behavior of POI gets over-represented or under-represented in fixed cell images. The intrinsic rates by which POI binds to and dissociates from its puncta impact the magnitude of fixation artifacts by determining both the live-cell equilibrium of LLPS (starting punctate percentage) and the relative overall fixation rate of POI ($(k_3 + k_4) : (k_1 + k_2)$).

## A fast overall fixation rate relative to binding dynamics can minimize fixation artifacts

As discussed above, our model suggests that when the overall fixation rate is fast compared with the dynamics of targeted protein–protein interactions, fixation artifacts can be minimized even with unequal fixation rates in and out of puncta. In order to test this prediction experimentally, we focused on Halo-TAF15(IDR), which exhibits significantly diminished LLPS behavior upon fixation (*Figure 2C*), and TAF15(IDR)-Halo-FTH1, which does not exhibit a significant fixation artifact (*Figure 3B*). The fact that fixation of both Halo-TAF15(IDR) and TAF15(IDR)-Halo-FTH1 are completed within 1–2 min suggests comparable overall fixation rates of the two proteins. Thus, our model predicts that TAF15(IDR)-Halo-FTH1 has more stable homotypic interactions than Halo-TAF15(IDR), resulting in a higher relative overall fixation rate of the former than the latter. To test this prediction, we performed live-cell single-molecule imaging of Halo-TAF15(IDR) and TAF15(IDR)-Halo-FTH1 (*Video 2*) and measured their binding residence times (RTs) at respective droplet-like puncta. Using established single-particle tracking (SPT) analysis (*Chong et al., 2018*), we found the RTs of TAF15(IDR) and TAF15(IDR)-FTH1 to be 10.23 ± 1.10 and 64.15 ± 11.65 s, respectively (*Figure 6*, *Figure 6—source data 1*). This result suggests significantly more stable binding of TAF15(IDR)-FTH1 than TAF15(IDR). Together, these imaging data are consistent with our model's prediction that a fast overall fixation rate relative to binding dynamics can minimize fixation artifacts.

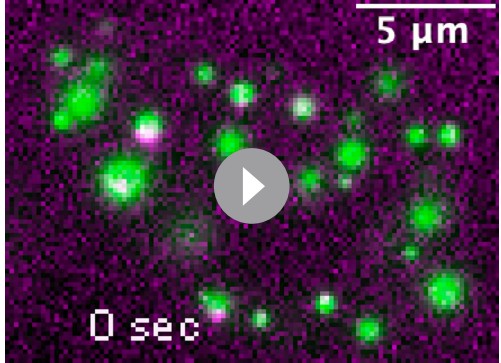

**Video 2.** A two-color real-time movie of individual molecules of TAF15(IDR)-Halo-FTH1 binding to its puncta.

https://elifesciences.org/articles/79903/figures#video2

## Discussion

Understanding situations in which PFA fixation can properly preserve live-cell conditions is essential in judicious applications of fixation-based biological techniques. Because approaches for rigorous determination of LLPS in vivo remain lacking (*McSwiggen et al., 2019b*) and detection of local high-concentration regions of an endogenously expressed protein in fixed cells via immunofluorescence has been widely used as evidence for LLPS (*Boija et al., 2018*; *Guo et al., 2019*; *Owen et al., 2021*; *Xie et al., 2022*; *Yang et al., 2020*), understanding how well fixation preserves LLPS behaviors is important for justifying the immunofluorescence-based diagnosis

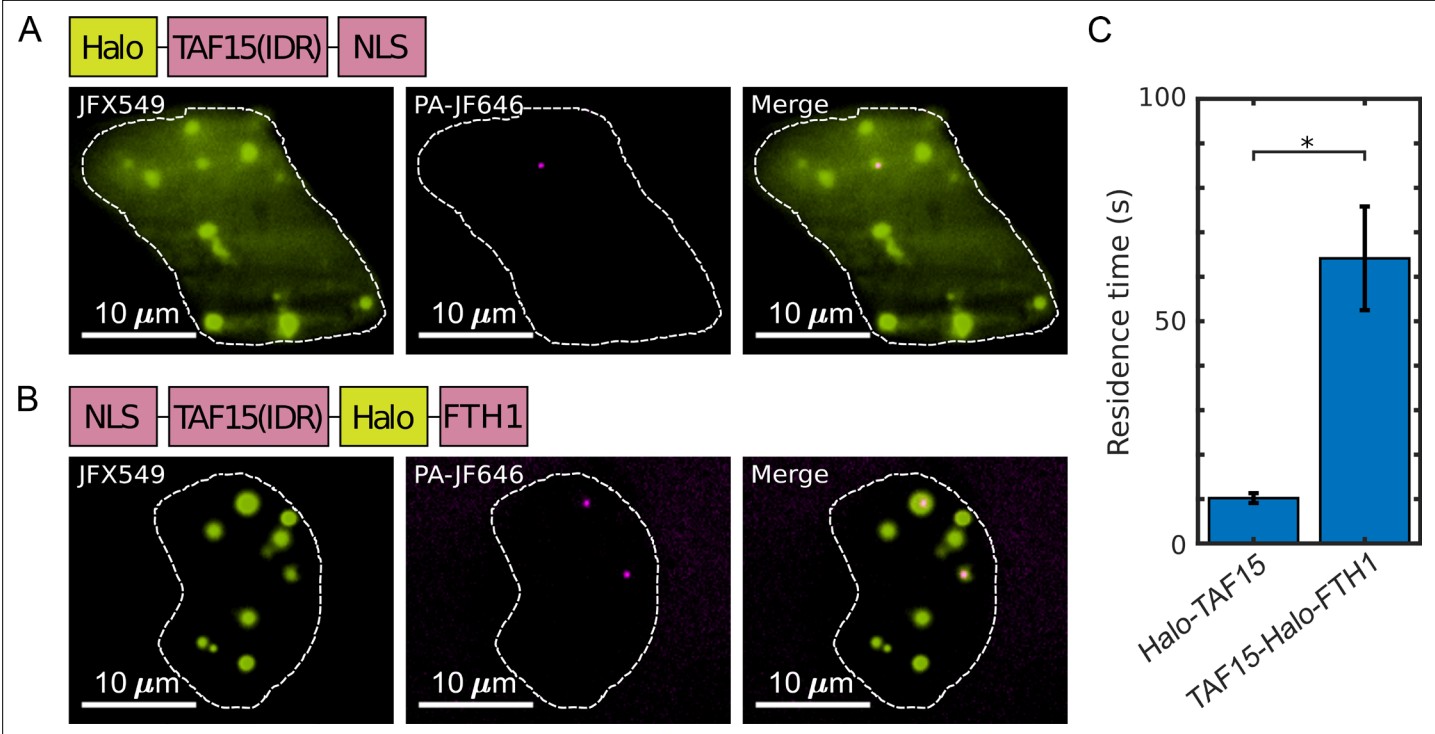

**Figure 6.** The residence times of proteins in their droplet-like puncta vary. Shown are individual frames from two-color single-molecule movies of (**A**) Halo-TAF15(IDR) and (**B**) TAF15(IDR)-Halo-FTH1. Each protein was labeled with a lower concentration of a photoactivatable dye for SPT (20 nM PA-JF646, magenta) and a higher concentration of non-photoactivatable dye for visualization of the droplet-like puncta (100 nM JFX549, yellow). A white dashed line outlines the nucleus. (**C**) The mean residence time of TAF15(IDR)-Halo-FTH1 in its puncta is significantly longer than that of Halo-TAF15(IDR) in its puncta. The value for each protein is averaged from 20 cells measured in three independent transfection and imaging sessions. Error bars represent standard errors. Asterisk indicates a significant difference between the two proteins ($p < 0.05$, Wilcoxon rank-sum test).

The online version of this article includes the following source data for figure 6:

**Source data 1.** Residence times (RTs) measured by single-particle tracking (SPT) used to generate the bar plots.

method and for studying the functional relevance of LLPS in vivo. In this work, we imaged various LLPS systems in living cells before and after PFA fixation, quantified parameters that describe LLPS appearance in cells, and showed that fixation can either enhance or diminish the apparent LLPS behaviors in vivo. Lowering the PFA concentration and adding GA to PFA did not remove the fixation artifacts. For the first time, our work reveals an important caveat in using fixation-based methods to detect and characterize LLPS in vivo and suggests an advantage of using live-cell imaging to study LLPS systems over fixed-cell experiments. However, not all the proteins we examined have their puncta-forming or apparent LLPS behaviors in cells changed upon fixation. For example, PFA fixation faithfully preserves the appearance of FUS(FL), TAF15(IDR)-FTH1, and EWS::FLI1 puncta in cells (*Figure 3*). Nevertheless, our work points out a necessity to use live-cell imaging to confirm LLPS behaviors previously characterized with fixed-cell experiments. Live-imaging techniques that allow estimation of protein diffusion coefficients within specific cellular compartments, for example, SPT (*Hansen et al., 2018*; *Heckert et al., 2022*) and fluorescence correlation spectroscopy (*Lanzanò et al., 2017*), can be useful alternative approaches for diagnosing LLPS in vivo without the potential artifact of fixation, as diffusion dynamics are recently shown to be affected by LLPS (*Heltberg et al., 2021*; *McSwiggen et al., 2019a*; *Miné-Hattab et al., 2021*; *Chong et al., 2022*; *Ladouceur et al., 2020*).

We note that fixation-induced changes of LLPS appearance may lead to potential misinterpretation of the functional relevance of LLPS in cellular processes. For example, recent work has uncovered that effective transcriptional activation requires an optimum of TF IDR-IDR interactions within TF hubs formed at target genes and that overly high levels of IDR-IDR interactions pushing the system toward LLPS can repress transcription (*Chong et al., 2022*; *Trojanowski et al., 2022*).

Future characterization of the functionally optimal interaction level will require quantification of the sizes of hubs or droplet-like puncta while measuring transcription activity. Because a fixation-induced increase or decrease in puncta sizes may lead to inaccurate determination of the functional optimum, scrutiny will be required in choosing between live-cell and fixed-cell imaging methods for quantifying LLPS appearance in these types of studies. Moreover, given that fixation can artificially generate intranuclear puncta of EGFP-EWS(IDR) that is homogenously distributed across the live-cell nucleus (*Figure 1A*), extra caution is required in interpreting immunofluorescence-detected intracellular puncta of an endogenously expressed protein as LLPS as the same puncta-generating fixation artifact might happen to the protein even when it is not phase separating in living cells. To confirm puncta formation, counterpart live-cell imaging of the endogenous protein will be necessary, which requires engineering the cells, for example, by CRISPR, to fluorescently tag the protein.

To understand the factors that can cause fixation-induced changes of LLPS appearance in the cell, we simulated the changes through kinetic modeling, which reveals that the dynamics of POI binding to and dissociating from puncta, the absolute fixation rate of POI, and different fixation rates of POI in and out of puncta all play a role in inducing the fixation artifacts. Our kinetic model takes previous work studying fixation artifacts in the context of protein–DNA interactions (*Poorey et al., 2013*; *Schmiedeberg et al., 2009*; *Teves et al., 2016*) one step further by considering two fixed states of POI instead of one state, which are fixation in and out of puncta with different rates due to distinct local protein composition and concentrations. We then used live-cell single-molecule imaging experiments to demonstrate that as predicted by our model, a fast overall fixation rate of POI relative to its puncta-binding dynamics can minimize fixation artifacts.

We emphasize that because our four-state model makes no assumptions about any state being phase-separated, the logical implications of our model can extend beyond LLPS to other biomolecular transactions and cellular structures that have been found not well preserved by fixation or immunofluorescence, including localizations of cilia proteins (*Hua and Ferland, 2017*), clustering of cell membrane receptors (*Stanly et al., 2016*), splicing speckle formation (*Neugebauer and Roth, 1997*), and chromatin organization and protein binding (*Zarębski et al., 2021*; *Lorber and Volk, 2022*; *Lerner et al., 2016*; *Pallier et al., 2003*; *Kumar et al., 2008* and *Teves et al., 2016*). Our model can similarly extend beyond PFA to other fixatives. This is useful because different fixatives have been chosen for studying different types of structures. For example, PFA fixation is often preferable for preserving soluble proteins over dehydration fixatives such as methanol (*Stadler et al., 2010* and *Schnell et al., 2012*), yet methanol fixation can be preferable over PFA for preserving proteins bound to mitotic chromatin (*Kumar et al., 2008*; *Lerner et al., 2016*). Generally, our model predicts that fixation artifacts will occur whenever a protein can exist in multiple states that have different rates of fixation, and this artifact is most severe when the fixation is slower than the transition between states. For PFA fixation, because its rate is sensitively dependent on the amino acid sequence of POI, the structure of POI, and POI's cross-linked partners (*Hoffman et al., 2015*; *Kamps et al., 2019*; *Metz et al., 2006*; *Metz et al., 2004*), POI in different states likely has different PFA fixation rates regardless of the type of interaction it undergoes.

One distinction between our study and previous studies is that we observe that PFA fixation can enhance apparent protein–protein interactions or LLPS behaviors in the cell, suggesting faster fixation for POI in the bound than dissociated state ($k_3 > k_4$), whereas fixation has only been reported to diminish protein–DNA interactions, suggesting slower fixation for POI in the bound state ($k_3 < k_4$) (*Poorey et al., 2013*; *Schmiedeberg et al., 2009*; *Teves et al., 2016*). We hypothesize that this is because fixing the bound state of an LLPS system (within puncta) is dominated by cross-linking reactions between IDRs enriched in puncta, which have reactive residues better exposed to solvent due to lack of well-defined tertiary structures and thereby likely cross-link faster than structured domains cross-linking to DNA (*Hoffman et al., 2015*). It will be of future interest to measure fixation rates of different biomolecules including IDRs, structured proteins, and nucleic acids to prove the proposed chemical mechanism underlying fixation artifacts. Since our simulated results highlight the role of absolute fixation rates in the outcome of fixation, another future endeavor will be

to design novel fixatives with significantly faster cross-linking rates than biomolecular interactions to eliminate fixation artifacts in the cell.

## Materials and methods

**Key resources table**

| Reagent type (species) or resource | Designation | Source or reference | Identifiers | Additional information |
|---|---|---|---|---|
| Cell line (human) | Knock-in A673 cell line | *Chong et al., 2018* | N/A | Human: A673 carrying HaloTag knock-in at the ews::fli1 locus |
| Cell line (human) | U2OS cell line | *Chong et al., 2018* | N/A | N/A |
| Recombinant DNA reagent | EGFP-EWS(IDR)-NLS | This paper | N/A | Plasmid encoding the protein See materials availability statement |
| Recombinant DNA reagent | EGFP-FUS(IDR)-NLS | This paper | N/A | Plasmid encoding the protein See materials availability statement |
| Recombinant DNA reagent | EGFP-TAF15(IDR)-NLS | This paper | N/A | Plasmid encoding the protein See materials availability statement |
| Recombinant DNA reagent | DsRed2-TAF15(IDR)-NLS | This paper | N/A | Plasmid encoding the protein See materials availability statement |
| Recombinant DNA reagent | Halo-TAF15(IDR)-NLS | *Chong et al., 2018* | N/A | Plasmid encoding the protein See materials availability statement |
| Recombinant DNA reagent | EGFP-FUS(FL) | This paper | N/A | Plasmid encoding the protein See materials availability statement |
| Recombinant DNA reagent | NLS-TAF15(IDR)-Halo-FTH1 | This paper | N/A | Plasmid encoding the protein See materials availability statement |
| Chemical compound, drug | Glycine | Fisher Scientific | Fischer Scientific: BP381-5 | N/A |
| Chemical compound, drug | Paraformaldehyde | VWR | VWR: 100503-917 | N/A |
| Chemical compound, drug | Glutaraldehyde | Sigma-Aldrich | Sigma-Aldrich: 340855-25ML | N/A |

### Cell line and sample preparation

U2OS cells were grown in 1 g/L DMEM media (Thermo Fisher, 10567014) supplemented with 10% FBS (Fisher Scientific, SH3039603) and 1% penicillin-streptomycin (Thermo Fisher, 15140122). The cells were split onto an imaging plate (Mattek, P35G-1.5-14-C) and transfected with fluorescent protein constructs with Lipofectamine 3000 (Fisher Scientific, L3000001) according to manufacturer's instructions. One day after transfection, the culture media was changed to phenol-red-free DMEM (Thermo Fisher, 11054001) with 10% FBS and 1% penicillin-streptomycin. For experiments with additional glycine, glycine (Fisher Scientific, BP381-5) was added to the phenol red-free media so that the final concentration was 50 mM (and 25 mM after the addition of 8% PFA, see below). It should be noted that normal DMEM media already contains 0.4 mM glycine. The knock-in A673 cell line expressing endogenous EWS::FLI1-Halo (*Chong et al., 2018*) was grown in 4.5 g/L DMEM media (Thermo Fisher, 10566016) with 10% FBS (Fisher Scientific, SH3039603) and 1% penicillin-streptomycin (Thermo Fisher, 15140122). The cells were similarly split onto an imaging plate (Mattek, P35G-1.5-14-C) and the culture media was changed to phenol-red-free DMEM (Thermo Fisher, 31053028) just before imaging. The U2OS cell line used here was validated by whole-genome sequencing as described in *Hansen et al., 2017*. The knock-in A673 cell line was generated by genome editing of the A673 cell line that was comprehensively authenticated by ATCC before distribution (ATCC, CRL-1598). The genomic sequence of the locus encoding EWS::FLI1-Halo in the knock-in A673 cell line was confirmed by Sanger sequencing. The knock-in A673 cell line was further authenticated using Short Tandem Repeat (STR) profiling (ATCC Cell Line Authentication Service) against the following loci: TH01, D5S818, D13S317, D7S820, D16S539, CSF1PO, Amelogenin, vWA, and TPOX. The knock-in A673

cell line showed a 100% match with A673. Both U2OS and A673 cell lines were tested for mycoplasma using PCR-based assays in February 2022.

## Fluorescence microscopy

Confocal fluorescence microscopy was performed on Zeiss LSM 980 in the point-scanning mode with a ×63 oil objective (Zeiss, 421782-9900-000). The pinhole was set to 1 airy unit for different emission wavelengths. The images displayed in the article are maximum z-projections of z-stack images. A673 cell expressing endogenous EWS::FLI1-Halo were imaged in the Airyscan mode of the same Zeiss LSM 980 microscope. All postprocessing parameters in the Airyscan analysis module were kept constant to guarantee a fair comparison between the images taken before and after fixation. The culture dish contained 1 mL of phenol red-free media, so that when 1 mL of 8% PFA (VWR, 100503-917) in PBS buffer was added to the dish, the final concentration of PFA was 4%. To achieve final PFA concentrations of 1, 2, and 8%, 1 mL of 2, 4, and 16% of PFA were diluted in PBS buffer and added to the culture dishes containing 1 mL of phenol red-free media. A final concentration of 0% was achieved by following the same protocol only using 1 mL of PBS buffer in place of PFA. To achieve final concentration of 4% PFA with 0.2% GA (Sigma-Aldrich, 340855-25ML), 1 mL of 8% PFA with 0.4% GA in PBS buffer was added to the culture dishes. After waiting 10 min to allow PFA or PFA/GA fixation to complete, images of the same cells are taken again. For experiments performed with glycine, cells were fixed with a final concentration of 4% PFA and 25 mM glycine. Independent transfection and imaging sessions were performed on different days using different plates of cells.

## LLPS parameter quantification

The three parameters we quantified were the number of puncta, surface roughness, and punctate percentage. The source code used to analyze the images is provided as a supplementary file 'Puncta Quantification Processing Scripts.zip.' To best compare the images of a cell before and after fixation, the two z-projection images were normalized so that the sum of the intensities within the nucleus is

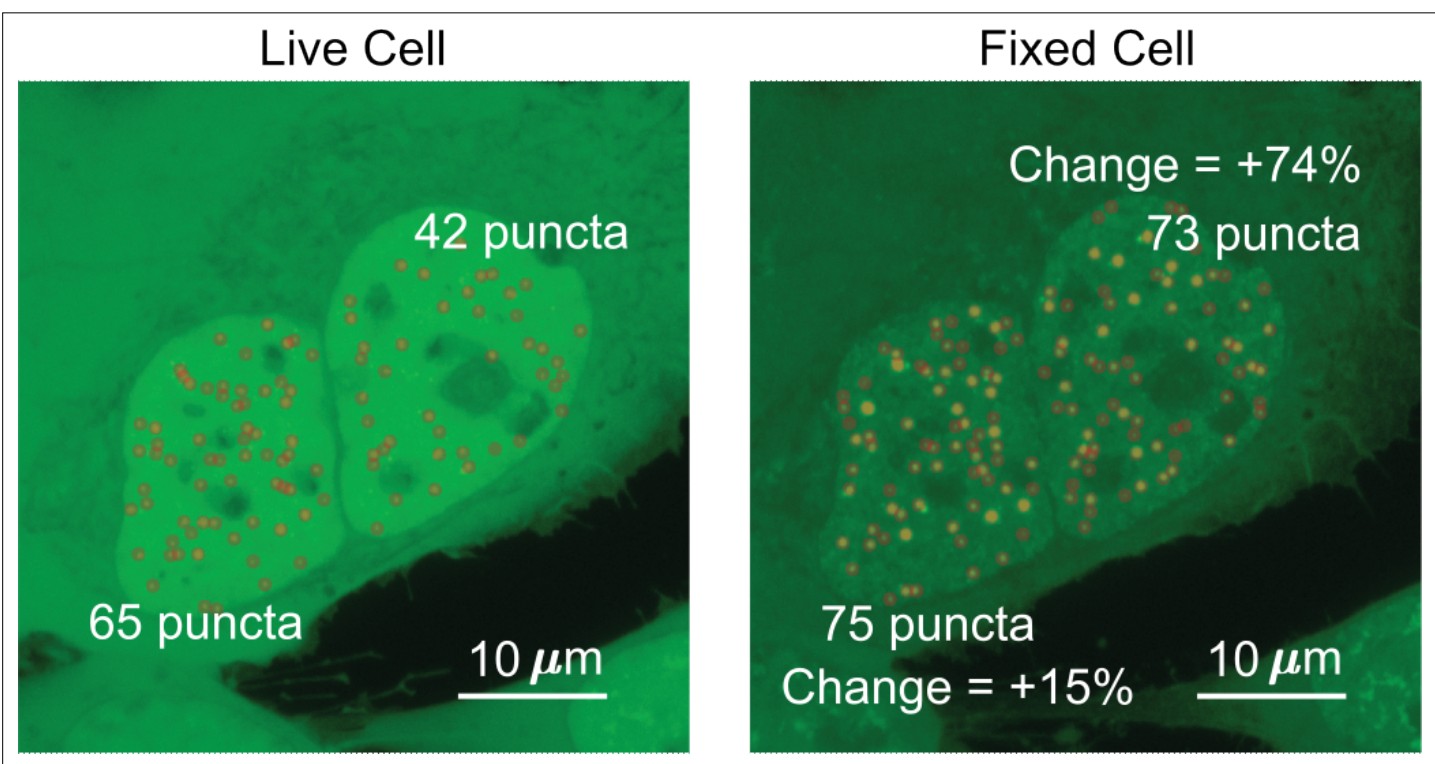

**Figure 7.** Determination of the number of puncta in the cell nucleus. Two cells expressing EGFP-TAF15(IDR) have the number of puncta before and after fixation compared. The cell on the left shows an increase of 10 puncta, a change of 15%. The cell on the right shows an increase of 31 puncta, a change of 74%.

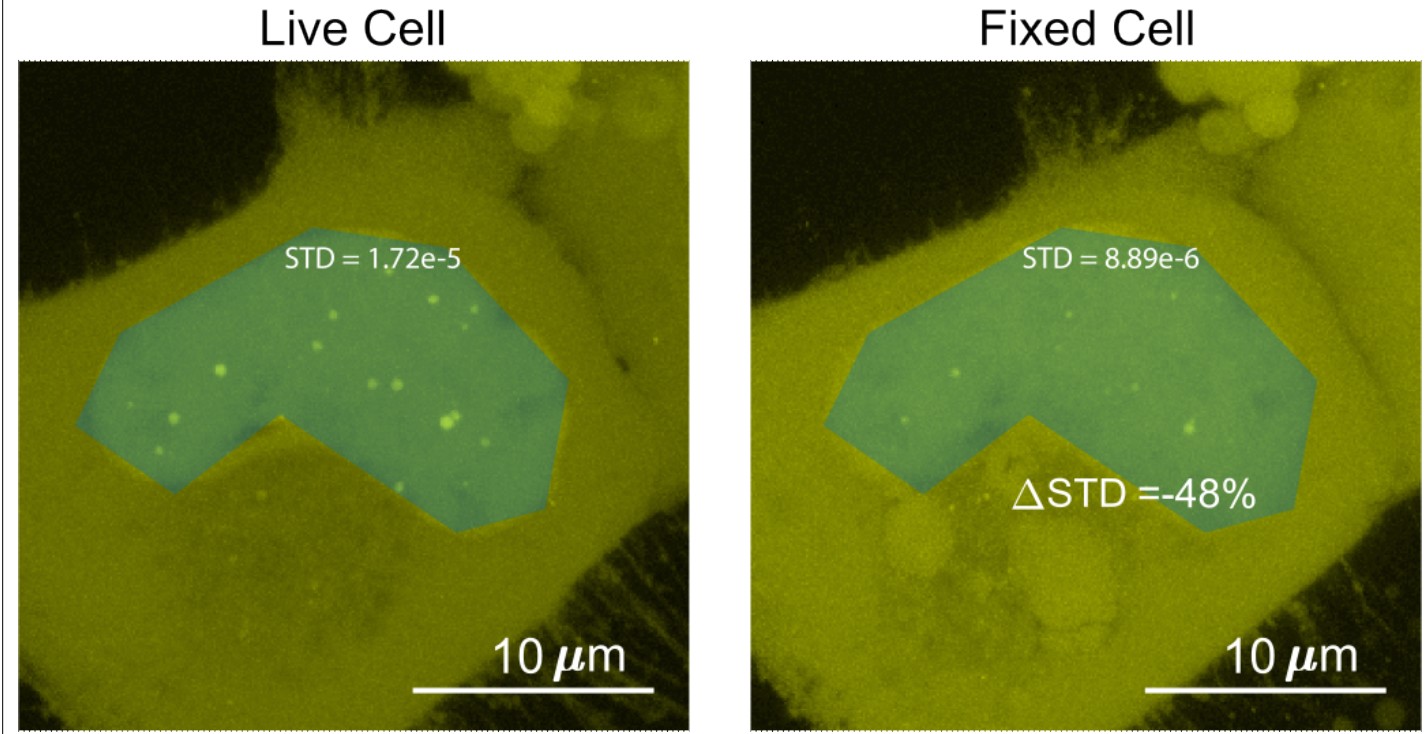

**Figure 8.** Determination of the surface roughness of a cell nucleus image. We drew a blue patch that covers the nucleus of a cell expressing Halo-TAF15(IDR) and compared the standard deviation of the pixel intensity within the blue patch before and after fixation. The change in standard deviation between the two images is –48%.

equal to 1. The border of the nucleus was manually drawn for each image. All analyses were done on normalized maximum z-projection images except for when calculating punctate percentage. We measured the number of puncta by quantifying the number of peaks within the nucleus. Specifically, the image was exported from MATLAB into ImageJ (*Schindelin et al., 2012*) using MIJ (*Sage et al., 2012*), and the 'find maxima' processing function was used (*Figure 7*) with the same noise tolerance for both live- and fixed-cell images.

To quantify the surface roughness of a cell nucleus image, the standard deviation of fluorescence intensities in the nucleus were compared before and after fixation (*Figure 8*). Utilizing this method of comparing images before and after fixation allows for quantification of change of nucleoplasm without peak fitting. The addition of structures such as puncta within a chosen patch will increase the standard deviation. Nuclei with puncta resulted in skewed (non-normal) distributions of intensities (*Jachowicz et al., 2021*), leading to higher standard deviations.

The punctate percentage was determined with the first few steps identical to measuring the number of puncta as described above. The border of the nucleus was manually identified, the images were normalized, and preliminary peak locations were identified on maximum z-projection images using the 'find maxima' function in ImageJ. The 'find maxima' function does not pick the perfect center of each punctum. Thus, to measure the full width at half maximum (FWHM) of a punctum, we made 36 different radial slices of the punctum crossing the preliminary punctum center pixel, extracted the intensity profile for each radial slice to calculate the punctum's FWHM, and selected the highest FWHM as the corresponding radial slice must have gone through the true center of the punctum. We then made a sum z-projection of the z-stack images, drew a circle with the maximum FWHM as its diameter centering the true central pixel of each punctum on the sum image, and integrated the fluorescence intensity across all circles (*Figure 9*). The punctate percentage is calculated by dividing the in-circle total fluorescence intensity with the total fluorescence intensity integrated across the nucleus in the sum image.

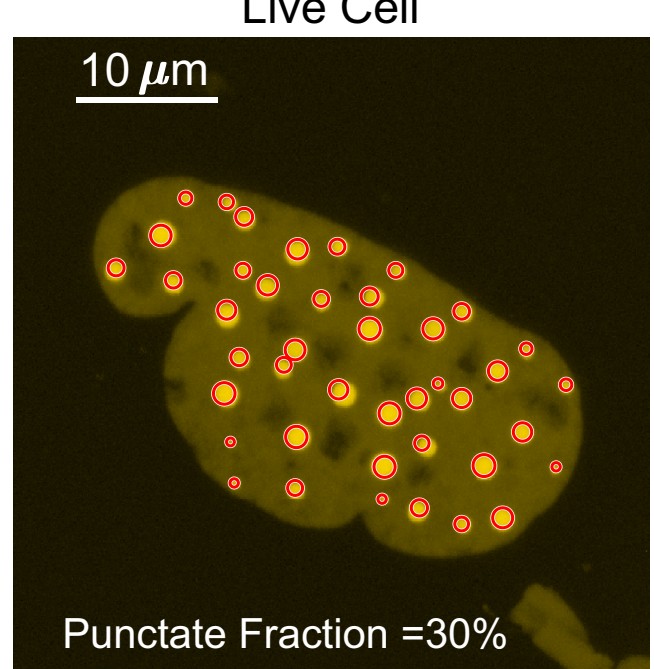

**Figure 9.** Determination of the punctate percentage. The punctate percentage of DsRed2-TAF15(IDR) is compared before and after fixation. The red circles represent the boundary within which the integrated fluorescence is considered 'in puncta'.

### Kinetic simulation

A four-state kinetic model was constructed in COPASI (*Hoops et al., 2006*) and interfaced using Python. The complete iPython notebook containing the source code used to perform the simulation is provided a supplementary file 'Kinetic Simulation.zip.' The four states and kinetic rates are defined in the main text and *Figure 5B*. We assume a constant total molarity for all species, that is, $[S_1] + [S_2] + [S_3] + [S_4] = 1 \, mol/L$. At $t = 0, [S_3] = [S_4] = 0$, while $k_1$ and $k_2$ together define the equilibrium between $S_1$ and $S_2$, that is, $K_{eq} = k_1/k_2 = [S_1]_{eq} / [S_2]_{eq}$. COPASI numerically simulates the four states in the kinetic model utilizing the starting concentrations and rate conditions.

The units used for all the rates were $s^{-1}$, set so that fixation occurred on the order of seconds. For the simulations that produced *Figure 5C*, we varied the values of $k_3$ and $k_4$ but kept the total fixation and POI binding and dissociation rates constant ($k_3 + k_4 = 0.2$, $k_1 + k_2 = 1$), leading to a constant relative overall fixation rate of POI ($(k_3 + k_4) : (k_1 + k_2)$=1:5). For the simulations that produced *Figure 5D*, we kept the fixation rates constant ($k_3 = 1$, $k_4 = 2$) and varied the relative overall fixation rate of POI ($(k_3 + k_4) : (k_1 + k_2)$). In this simulation, the relative overall fixation rate of POI ($(k_3 + k_4) : (k_1 + k_2)$) is set so that the range of interaction rates span values that are an order of magnitude faster and slower than fixation rates.

### Single-particle tracking (SPT)

SPT of Halo-tagged TAF15(IDR) and TAF15(IDR)-FTH1 were performed on a Nikon Eclipse Ti2 TIRF microscope with a ×100/NA 1.49 oil-immersion objective (CFI SR HP Apochromat TIRF 100XAC Oil) under highly inclined and laminated optical sheet (HILO) illumination (*Tokunaga et al., 2008*). PA-JF646 was activated and excited under variable powers by 405 nm and 640 nm laser lines, respectively, while JFX549 was excited by a 561 nm laser line. The incubation chamber was held humidified at a 37°C with 5% $CO_2$ and the objective was also heated to 37°C.

Halo-tagged TAF15(IDR) and TAF15(IDR)-FTH1 were overexpressed in U2OS cells and stained with 100 nM JFX549 (*Grimm et al., 2021*) and 20 nM PA-JF646 (*Grimm et al., 2016*). Droplet-like puncta

were visualized in the JFX549 channel, while individual molecules detected in the PA-JF646 channel were tracked in real time. A low 405 nm activation power was used to ensure sufficiently sparse activation of PA-JF646-labeled proteins and allow for SPT. For SPT in the PA-JF646 channel, long camera exposure time (500 ms per frame, 2000 frames) blurred out faster diffusing molecules and ensured that we only detect bound molecules. In the JFX549 channel, time-lapse images (500 ms per frame, one frame every 10 s) were taken to track the location of droplet-like puncta during the entire acquisition while limiting the effects of photobleaching.

The analysis was performed following *Chong et al., 2018* and is briefly described below. Single-molecule data from the PA-JF646 channel was analyzed using a SLIMfast (*Normanno et al., 2015*), a GUI based on a MATLAB implementation of the MTT algorithm (*Sergé et al., 2008*), and is available in the supplemental materials of *Teves et al., 2016*. SPT analysis was performed using the following parameters: localization error: $10^{-6}$; deflation loops: 3; maximum number of competitors: 5; maximal expected diffusion constant ($\mu m^2/s$): 0.1.

Binary masks of the droplet-like puncta were generated from the JFX549 channel using custom-written Macros in ImageJ from *Chong et al., 2018*. Using custom-written MATLAB code also from *Chong et al., 2018*, single-molecule trajectories were then sorted into in-puncta and out-of-puncta trajectories based on the fraction of time a molecule spent in the punctum, F. A trajectory with $F > 50\%$ was considered in-puncta and one with $F < 5\%$ was considered out of puncta. We only focused on the in-puncta trajectories.

Survival probability curves were then generated from the in-puncta trajectories and fit to the following two-component exponential model.

$$P\left(t\right) = Ae^{-k_1 t} + \left(1 - A\right)e^{-k_2 t},$$
$$1/k_1 = \tau_{ns}, 1/k_2 = \tau_s, \tag{2}$$

with $\tau_{ns}$ and $\tau_s$ as the specific and nonspecific residence times, respectively. Here, we only focused on the specific residence times.

In order to correct for photobleaching (*Hansen et al., 2017*), the specific residence time of histone H2B (which is largely immobile on the chromatin) was measured via SPT as described above, except on all trajectories rather than doing the in-puncta and out-of-puncta classification. We used PA-JF646-tagged H2B-Halo that was stably expressed in U2OS cells and imaged under illumination and acquisition parameters identical to those used to image Halo-tagged TAF15 and TAF15-FTH1. The corrected specific residence times of the Halo-tagged TAF15 and TAF15-FTH1 ($\tau_{\text{corrected}}$) were computed based on the following model.

$$\tau_{\text{corrected}} = 1/\left(1/\tau_s - 1/\tau_{\text{H2B}}\right), \tag{3}$$

with $\tau_{H2B}$ as the specific residence time of H2B.

Independent experiments were performed across at least three days for both Halo-tagged TAF15 and TAF15-FTH1. In each session, multiple movies of both constructs were taken along with three movies of Halo-tagged H2B to perform correction for that specific day. We reported the mean corrected residence times.

## Statistical analysis

Nonparametric tests used throughout because the data were often not normally distributed. Statistical significance of the LLPS parameters was calculated using the Wilcoxon signed-rank test and statistical significance of the residence times from SPT was using the Wilcoxon signed-rank test (*Gibbons and Chakraborti, 2014*). The Wilcoxon signed-rank test and Wilcoxon rank-sum test were performed using the built-in MATLAB functions *signrank* and *ranksum*, respectively.

## Acknowledgements

This work was supported by the Shurl and Kay Curci Foundation Research Grant (to S Chong), the John D Baldeschwieler and Marlene R Konnar Foundation (to S Chong), Pew-Stewart Scholars Program for Cancer Research (to S Chong), Searle Scholars Program (to S Chong), and Merkin Innovation

Seed Grant (to S Chong). We thank Luke Lavis for providing fluorescent HaloTag ligands; the Caltech Biological Imaging Facility and Giada Spigolon for providing technical assistance on confocal fluorescence microscopy; and Robert Tjian, Thomas Graham, John Ferrie, and Jonathan Karr for critical comments on the manuscript.

## Additional information

### Funding

| Funder | Grant reference number | Author |
| --- | --- | --- |
| Shurl and Key Curci Foundation | Research Grant | Shasha Chong |
| John D. Baldeschwieler and Marlene R. Konnar Foundation | | Shasha Chong |
| Pew-Stewart Scholars Program for Cancer Research | | Shasha Chong |
| Searle Scholars Program | | Shasha Chong |
| Merkin Institute for Translational Research | Merkin Innovation Seed Grant | Shasha Chong |

The funders had no role in study design, data collection and interpretation, or the decision to submit the work for publication.

### Author contributions

Shawn Irgen-Gioro, Data curation, Software, Formal analysis, Validation, Investigation, Visualization, Methodology, Writing – original draft, Writing – review and editing; Shawn Yoshida, Data curation, Formal analysis, Validation, Investigation, Visualization, Methodology, Writing – review and editing, Writing – original draft; Victoria Walling, Investigation; Shasha Chong, Conceptualization, Resources, Supervision, Funding acquisition, Validation, Investigation, Visualization, Methodology, Writing – original draft, Project administration, Writing – review and editing

### Author ORCIDs

Shawn Irgen-Gioro http://orcid.org/0000-0001-8638-6191
Shawn Yoshida http://orcid.org/0000-0002-0866-2741
Shasha Chong http://orcid.org/0000-0002-5372-311X

### Decision letter and Author response

Decision letter https://doi.org/10.7554/eLife.79903.sa1
Author response https://doi.org/10.7554/eLife.79903.sa2

## Additional files

### Supplementary files

- MDAR checklist
- Source code 1. Code used to quantify number of puncta, surface roughness of cell nucleus, and punctate percentage.
- Source code 2. Code used to perform kinetic simulation of the four-state fixation model.

### Data availability

Figure 1 - Source Data 1, Figure 2 - Source Data 1, Figure 3 - Source Data 1, and Figure 6 - Source Data 1 contain the numerical data used to generate the figures. Custom scripts have been uploaded as source code files.

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
