## [Editor Report]

Chemically fixing cells for fluorescence microscopy is a common practice in cell biology. However, fixation artifacts can lead the incorrect interpretations of experimental results. This article presents compelling evidence showing that in the context of liquid condensates formed by liquid–liquid phase separation (LLPS), paraformaldehyde (PFA) fixation creates a number of artifacts – such as changes in the number, appearance, or disappearance of liquid condensates. These important findings will be of great interest not only for those in the LLPS field but for any cell biologists using fixed samples for microscopy.

---

## [Decision Letter]

**Decision letter after peer review:**

Thank you for submitting your article "Fixation Can Change the Appearance of Phase Separation in Living Cells" for consideration by *eLife*. Your article has been reviewed by 3 peer reviewers, including Felix Campelo as Reviewing Editor and Reviewer #1, and the evaluation has been overseen by Vivek Malhotra as the Senior Editor. The following individual involved in the review of your submission has agreed to reveal their identity: Judith Miné-Hattab (Reviewer #2).

Essential revisions:

In the discussion amongst the reviewers, we all agreed that this is a very interesting and important paper. But the paper needs some clarifications and extra work. Specifically, the major points that should be addressed follow.

1) Compare different fixation methods/concentrations, etc. The authors only report on the artifacts caused by PFA at 4% (with 10' incubation time). They should test other concentrations and other fixatives. In particular, work from the Eggeling lab showed a similar fixation artifact on cell surface receptor clustering (https://journals.biologists.com/bio/article/5/9/1343/1227/Critical-importance-of-appropriate-fixation), which was somehow minimized by adding glutaraldehyde to the fixation buffer. The authors should also test this.

2) Better relate the experimental observations with the dynamic model. As suggested (see detailed reports below), this could be by measuring diffusion coefficients (e.g. FRAP) and see if that matches with the model's predictions.

3) Improve the description of LLPS in the introduction (see details below).

4) Finally, although we do not require the authors to test how other cellular structures are affected by fixation, the authors should add a short section in the introduction to mention some examples of other kinds of structures that are not well conserved by the fixation. Otherwise, it gives the impression that fixation artifacts exist only for LLPS, unfortunately, the preservation of structures by fixation is not limited to LLPS. They should also discuss the fact that some fixation protocols can destroy some structures while another protocol will preserve them. It is the case for some filament structures such as actin filament which are preserved by methanol fixation but seem altered by PFA fixation. This part would help the reader to understand that the quality of a fixation protocol strongly depends on the type of structure studied.

*Reviewer #1 (Recommendations for the authors):*

I think this is an important paper that presents an important observation that will be important for the community. My main concern/question is whether the two parts of the paper (experimental observations and computational model) are connected causally or not. I think that the glycine experiments point in that direction, but I fail to see concluding evidence on whether the observed changes in the LLPS structures after fixation are indeed caused by slower/faster fixation rates in/out of the condensates. Along these lines:

– Line 111: "The fact that different phase-separating proteins can have bifurcating behaviors upon fixation is interesting.": I fully agree with this. Have the authors considered coexpressing in the same cells the same protein with two different tags (that behave differently after fixation), such as GFP-TAF15 and dsRed-TAF15; or probably even better, dsRED and Halo TAF15? If the kinetic model represents the experimental situation, wouldn't you expect that fixation leads still to the appearance of small droplets in the dsRED but not in the Halo protein?

– Figure 4: glycine also seems to cause a change in the fraction of droplet protein in live cells (compare left panels in A and B). Could the authors discuss that?

Related to the model:

– Line 154: k2 in the model appears as a volumetric rate (that is, all particles in the droplet have the same escape probability). Would a surface escape rate (only particles at the surface are able to escape) change the results of the model?

– Can the authors provide analytical expressions (I believe that is relatively simple) for the plots in 5C, D as a function of the relative in puncta fix. rate and the relative overall fix. rate?

*Reviewer #2 (Recommendations for the authors):*

To strengthen the manuscript, the authors should try more protocols of fixation. In the simulation part, they could try to incorporate the diffusion coefficient of the protein of interest and see if it is possible to predict the effect of fixation as a function of the diffusion coefficient.

The manuscript focuses on LLPS but it would be interesting to discuss other artefacts of fixation outside of the LLPS: have they tested the artefacts on structures like filaments, chromatin organization, or other types of structures than LLPS? Or do fixation artefacts only concern LLPS?

*Reviewer #3 (Recommendations for the authors):*

Proteins that undergo LLPS in living cells show a very dynamic behavior and rapidly move from the biomolecular condensates to the surrounding environment (e.g. cytoplasm or nucleoplasm), as demonstrated using FRAP. This dynamic behavior could explain why when comparing fluorescently tagged IDR proteins in living cells and in fixed cells, one could detect only the "larger" condensates in living cells, while detecting also smaller condensates in the fixed cells. Comparison of the number and size of condensates detected in living cells using conventional confocal microscopy and super-resolution microscopy will help understand whether this is sufficient to increase the number of smaller condensates detected in living cells. If not, this would support the idea proposed by the authors that "when the overall fixation rate is fast compared with the dynamics of targeted interactions, fixation artifacts can be minimized even with unequal fixation rates in and out of puncta."

Not all proteins analyzed showed a different distribution in living versus fixed cells (as shown in Figure 3 for GFP-FUS). The differences in the number and size of condensates observed in living and fixed cells should be correlated with the dynamic of the protein analyzed by FRAP. Are proteins with the highest mobility measured by FRAP corresponding to those that show an increased number/size of puncta upon fixation? Are FUS protein-protein interactions stable and less dynamic compared with the overall fixation rate?

Recommendations for improving the writing and presentation: Defining whether a protein undergoes LLPS is based on different assays in vitro, using recombinant proteins and in cells. The observation that a given protein forms "puncta" inside the cells is generally not accepted as a criterion to establish whether it undergoes LLPS. The measure of the number, size, sphericity, liquid-like dynamic behavior (e.g. by FRAP), and sensitivity to agents such as e.g. hexanediol are all assays required to establish and characterize whether a given protein undergoes LLPS. These aspects should be described in the introduction of the paper. It is a bit simplistic to only focus on the calculation of the number and size of puncta before/after fixation.

---

## [Author Response]

Essential revisions:In the discussion amongst the reviewers, we all agreed that this is a very interesting and important paper. But the paper needs some clarifications and extra work. Specifically, the major points that should be addressed follow.1) Compare different fixation methods/concentrations, etc. The authors only report on the artifacts caused by PFA at 4% (with 10' incubation time). They should test other concentrations and other fixatives. In particular, work from the Eggeling lab showed a similar fixation artifact on cell surface receptor clustering (https://journals.biologists.com/bio/article/5/9/1343/1227/Critical-importance-of-appropriate-fixation), which was somehow minimized by adding glutaraldehyde to the fixation buffer. The authors should also test this.

We thank the reviewers for this constructive comment and agree that various concentrations of PFA along with PFA supplemented by glutaraldehyde (GA) should be tested for potential fixation artifacts. Following the reviewer’s suggestions, we have performed all the recommended experiments and included new data in the revised manuscript to address this concern.

1) We quantified the artifact resulting from fixation using 0% PFA (PBS buffer only), 1% PFA, 2% PFA, and 8% PFA besides 4% PFA. We showed that all concentrations of PFA other than 0% resulted in statistically significant change in all the parameters that quantify LLPS appearance (“number of puncta”, “surface roughness”, and “punctate percentage”), though a quantitative comparison of parameter changes at different PFA concentrations is difficult due to increased fluorescence quenching effects at higher PFA concentrations. The description of our new data was added to the revised manuscript in Paragraph 3 on Page 3 and in Figure 1 —figure supplement 3.

2) We quantified the artifact resulting from fixation using 4% PFA supplemented with 0.2% GA, the same fixative used in (https://journals.biologists.com/bio/article/5/9/1343/1227/Critical-importance-of-appropriatefixation). We found a statistically significant change in all the LLPS-describing parameters. We also compared the fixation artifact resulting from PFA/GA in combination with the fixation artifact resulting from PFA alone and found no statistically significant differences. The description of our new data was added to the revised manuscript in Paragraph 3 on Page 3 and in Figure 1 —figure supplement 4.

Overall, these new results showed that fixation artifacts for LLPS systems are not unique to fixation with 4% PFA; rather, they can result from a broad range of PFA concentrations and cannot be eliminated by the addition of GA.

2) Better relate the experimental observations with the dynamic model. As suggested (see detailed reports below), this could be by measuring diffusion coefficients (e.g. FRAP) and see if that matches with the model's predictions.

We thank the reviewers for identifying this disconnect and agree that experiments probing the dynamics are needed to bridge the gap between our experimental observations and kinetic model of the fixation effects. Following the reviewers’ suggestions, we have now included a significant amount of new data in the revised manuscript to address this concern.

Our model does not predict the impact of diffusion dynamics on cell fixation, instead, it states that association rate, dissociation rate, in-puncta fixation rate, and out-of-puncta fixation rate together determine the fixation artifact. In particular, our model predicts that a fast overall fixation rate relative to binding dynamics can result in a reduced fixation artifact. In the revised manuscript, we have performed a series of new imaging experiments to prove this important prediction. The two proteins we focused on are Halo-TAF15(IDR) that experiences a significant fixation artifact and TAF15(IDR)-Halo-FTH1 that does not experience any fixation artifact (Figures 2C and 3B in the revised manuscript). Our model predicts that compared with Halo-TAF15(IDR), TAF15(IDR)-Halo-FTH1 has a faster overall fixation rate relative to binding dynamics. Since fixation of both proteins are completed within 1~2 minutes, it is predicted that TAF15(IDR)-Halo-FTH1 binds to its droplet-like puncta less dynamically than Halo-TAF15(IDR) does. We used single-particle tracking (SPT) to measure the residence times of Halo-TAF15(IDR) and TAF15(IDR)-Halo-FTH1 in respective puncta and showed that TAF15(IDR)-Halo-FTH1 binds to its puncta significantly longer than Halo-TAF15(IDR) (Figure 6 and Video 2 in the revised manuscript), suggesting significantly more stable homotypic interactions of the former than the latter and a higher relative overall fixation rate of the former than the latter. In short, our experimental observations are perfectly consistent with the model prediction.

The fixation artifact of TAF15(IDR)-Halo-FTH1 was described in Paragraph 1 on Page 8 and Figures 3B and 3E of the revised manuscript. The description of our SPT results and how they are related to our kinetic model were added to Paragraph 3 on Page 13, Figure 6, and Video 2. We have also updated Figure 5D to show the simulation of cell fixation with diminished rather than enhanced LLPS to be consistent with our observation for Halo-TAF15(IDR). This makes it easier for the readers to compare our experimental observations to our model simulations.

3) Improve the description of LLPS in the introduction (see details below).

We thank the reviewers for this suggestion and have updated the description of LLPS in the introduction in Paragraph 2 on Page 2 as follows:

“Whereas rigorous characterization of LLPS in vivo has been challenging and remains a question under active investigation (McSwiggen et al., 2019b), detection of discrete puncta that have a spherical shape, undergo fusion and fission, and dynamically exchange biomolecules with the surrounding according to FRAP is often considered evidence of putative LLPS in living cells. While such diverse measurements have been widely used for studying proteins under overexpression conditions, far fewer approaches are available to probe LLPS under physiological conditions. Detecting local high-concentration regions or puncta of an endogenously expressed protein using immunofluorescence of fixed cells has been used in many studies as evidence of LLPS ...”

4) Finally, although we do not require the authors to test how other cellular structures are affected by fixation, the authors should add a short section in the introduction to mention some examples of other kinds of structures that are not well conserved by the fixation. Otherwise, it gives the impression that fixation artifacts exist only for LLPS, unfortunately, the preservation of structures by fixation is not limited to LLPS. They should also discuss the fact that some fixation protocols can destroy some structures while another protocol will preserve them. It is the case for some filament structures such as actin filament which are preserved by methanol fixation but seem altered by PFA fixation. This part would help the reader to understand that the quality of a fixation protocol strongly depends on the type of structure studied.

We thank the reviewers for suggesting that we clarify that fixation artifacts are not exclusive to LLPS and that the quality of a fixation protocol strongly depends on the type of structure being studied. We have updated the Discussion section to address this concern in Paragraph 4 on Page 15:

“We emphasize that because our four-state model makes no assumptions about any state being phase-separated, the logical implications of our model can extend beyond LLPS to other biomolecular transactions and cellular structures that have been found not well preserved by fixation or immunofluorescence, including localizations of cilia proteins (Hua and Ferland, 2017), clustering of cell membrane receptors (Stanly et al., 2016), splicing speckle formation (Neugebauer and Roth, 1997), and chromatin organization and protein binding (Zarębski et al., 2021; Lorber and Volk, 2022, Lerner et al., 2016; Pallier et al., 2003; Kumar et al., 2007; and Teves et al., 2016). Our model can similarly extend beyond PFA to other fixatives. This is useful because different fixatives have been chosen for studying different types of structures. For example, PFA fixation is often preferable for preserving soluble proteins over dehydration fixatives such as methanol (Stadler et al., 2010 and Schnell et al., 2012), yet methanol fixation can be preferable over PFA for preserving proteins bound to mitotic chromatin (Kumar et al., 2007 and Lerner et al., 2016)”.

Reviewer #1 (Recommendations for the authors):I think this is an important paper that presents an important observation that will be important for the community. My main concern/question is whether the two parts of the paper (experimental observations and computational model) are connected causally or not. I think that the glycine experiments point in that direction, but I fail to see concluding evidence on whether the observed changes in the LLPS structures after fixation are indeed caused by slower/faster fixation rates in/out of the condensates.

We thank the reviewer for sharing this concern/question. We’d like to start by explaining the importance of our 4-state kinetic model involving fixed states in and out of puncta. While excellent previous works (Poorey et al., 2013; Schmiedeberg et al., 2009; and Teves et al., 2016) have proposed models that explain their respective observed fixation artifacts, their models are unable to capture how fixation has the capability to both enhance and diminish LLPS behavior that we observed experimentally. The inability of previous models to describe our experimental results motivated us to propose the 4-state kinetic model, which perfectly simulates the bifurcating behaviors. Although there is currently not a method capable of directly measuring fixation rates in and out of puncta in a cell, difference between the two fixation rates is well supported by well-established facts that (1) the dilute and concentrated phases of a LLPS system have different protein composition and concentrations (Currie and Rosen, 2022; Koga et al., 2011; Nott et al., 2015; Yewdall et al., 2021) and (2) fixation rate varies with both protein composition and concentration by orders of magnitude (Hoffman et al., 2015; Kamps et al., 2019; Metz et al., 2006; Metz et al., 2004). We have listed the facts in Paragraph 2 on Page 11 in the revised manuscript. Despite the infeasibility of measuring fixation rates in and out of puncta and showing experimentally the difference causes a fixation artifact, we have performed SPT experiments to measure the binding dynamics of different proteins within respective puncta and showed that a fast overall fixation rate relative to binding dynamics can minimize fixation artifacts, proving a different critical prediction by our model. We believe our new data successfully bridged the previous gap between experiments and the model. Please see Response 2 for the details of our additional experiments to address the concern.

Along these lines:– Line 111: "The fact that different phase‐separating proteins can have bifurcating behaviors upon fixation is interesting.": I fully agree with this. Have the authors considered coexpressing in the same cells the same protein with two different tags (that behave differently after fixation), such as GFP-TAF15 and dsRed-TAF15; or probably even better, dsRED and Halo TAF15? If the kinetic model represents the experimental situation, wouldn't you expect that fixation leads still to the appearance of small droplets in the dsRED but not in the Halo protein?

We thank the reviewer for this suggestion. Although we agree that this would make a very interesting experiment, this experimental design is unfortunately not feasible in practice. Because TAF15 undergoes strong homotypic interactions (Chong et al., 2018), TAF15 labeled with two different tags, e.g., dsRED and Halo, will colocalize in the same puncta instead of forming separate puncta each containing dsRED-TAF15 or Halo-TAF15 only. Such a homotypic interaction capability is ubiquitous for IDRs with a LLPS potential. Thus, our kinetic model can only account for one protein species in the cell at a time and the model is unsuitable to describe the fixation artifacts for the suggested case.

– Figure 4: glycine also seems to cause a change in the fraction of droplet protein in live cells (compare left panels in A and B). Could the authors discuss that?

We thank the reviewer for this suggestion and have added data and discussion in Paragraph 1 on Page 10 as follows:

“We found that adding 25mM glycine to live U2OS cells that overexpress DsRed2-TAF15(IDR) increases the starting punctate percentage from 18 ± 1.92 to 36 ± 3.82% (quantified from 23 cells), indicating an increase in the degree of LLPS. Although the underlying mechanism of such increase is unclear, we speculate this might be because hydrophobic intermolecular contacts that play an important role in TAF15(IDR) LLPS (Patel et al., 2017) are enhanced by the presence of hydrophobic glycine.”

Related to the model:– Line 154: k2 in the model appears as a volumetric rate (that is, all particles in the droplet have the same escape probability). Would a surface escape rate (only particles at the surface are able to escape) change the results of the model?

We thank the reviewer for this interesting question. Our model does not explicitly model the spatial distribution of proteins; rather, it only considers the average exchange rates between S1 and S2 at the population level. This means that rather than separately considering the potentially different dissociation rates of POI molecules on the surface and interior of a punctum, an ensemble dissociate rate of POI molecules from puncta is used here. An in-depth model that distinguishes between surface and interior dissociation rates could certainly provide further insight into the makeup of the ensemble dissociation rate but would ultimately not change the simulation results of our model.

We have added discussion clarifying this point in the revised manuscript in Paragraph 1 on Page 11:

“These are the average exchange rates between S1 and S2 and do not concern the potential spatial inhomogeneity in the rates at the molecular level. For example, individual POI molecules at the surface and interior of a punctum might dissociate with different rates, but our model does not differentiate these molecules.”

– Can the authors provide analytical expressions (I believe that is relatively simple) for the plots in 5C, D as a function of the relative in puncta fix. rate and the relative overall fix. rate?

We thank the reviewer for this suggestion. We have provided the analytical solutions to the differential equations posed by our model below. Since they turned out to be too cumbersome to provide additional insight to readers, we chose to numerically calculate our results in the manuscript as Figure 5C and 5D.

Here, we present analytical solutions to the following systems of differential equations.

{dS1dt=k1S2−k2S1−k3S1dS2dt=−k1S2+k2S1−k4S2dS3dt=k3S1dS4dt=k4S2

We have

{S1(t)=−C4k2k4(k1k3+k2k4+k3k4+(k2+k3)ξ+)etξ+−C3k2k4(k1k3+k2k4+k3k4+(k2+k3)ξ−)etξ−S2(t)=C4k4ξ+etξ++C3k4ξ−etξ−S3(t)=C1+C4k3k2k4(k1+k4+ξ+)etξ++C3k3k2k4(k1+k4+ξ−)etξ−S4(t)=C2+C4etξ++C3etξ−

Where C_1_, C_2_, C_3_ and C_4_ are constants dependent on the initial conditions and where

ξ±=±k12+2k1k2−2k1k3+2k1k4+k22+2k2k3−2k2k4+k32−2k3k4+k422−(k1+k2+k3+k4)2

Reviewer #2 (Recommendations for the authors):To strengthen the manuscript, the authors should try more protocols of fixation. In the simulation part, they could try to incorporate the diffusion coefficient of the protein of interest and see if it is possible to predict the effect of fixation as a function of the diffusion coefficient.The manuscript focuses on LLPS but it would be interesting to discuss other artefacts of fixation outside of the LLPS: have they tested the artefacts on structures like filaments, chromatin organization, or other types of structures than LLPS? Or do fixation artefacts only concern LLPS?

We thank the reviewer for raising this good point. This current work focuses on studying the effects of cell fixation on the appearance of LLPS. Although we do not investigate fixation artifacts of other cellular structures here, we have now discussed previously characterized fixation artifacts on other structures in the revised manuscript, e.g, localizations of cilia proteins (Hua and Ferland, 2017), clustering of cell membrane receptors (Stanly et al., 2016), splicing speckle formation (Neugebauer and Roth=, 1997), and chromatin organization and protein binding (Zarębski et al., 2021; Lorber and Volk, 2022, Lerner et al., 2016; Pallier et al., 2003; Kumar et al., 2007; and Teves et al., 2016). As also mentioned in Response 4, we have discussed this important point in Paragraph 4 on Page 15.

Reviewer #3 (Recommendations for the authors):Proteins that undergo LLPS in living cells show a very dynamic behavior and rapidly move from the biomolecular condensates to the surrounding environment (e.g. cytoplasm or nucleoplasm), as demonstrated using FRAP. This dynamic behavior could explain why when comparing fluorescently tagged IDR proteins in living cells and in fixed cells, one could detect only the "larger" condensates in living cells, while detecting also smaller condensates in the fixed cells. Comparison of the number and size of condensates detected in living cells using conventional confocal microscopy and super-resolution microscopy will help understand whether this is sufficient to increase the number of smaller condensates detected in living cells. If not, this would support the idea proposed by the authors that "when the overall fixation rate is fast compared with the dynamics of targeted interactions, fixation artifacts can be minimized even with unequal fixation rates in and out of puncta."

We appreciate the reviewer for this interesting idea and agree that it is possible that there are sub-diffraction limit puncta of DsRed2-TAF15(IDR) in cells, which were not detected using conventional confocal microscopy, and these small puncta potentially have more dynamic protein dynamics than the larger puncta and can potentially be affected by fixation more significantly. Although we agree that a comparison of DsRed2-TAF15(IDR) distribution under confocal and super-resolution microscopy would be a valuable set of experiments, performing super-resolution imaging of DsRed2 would require highly specialized equipment (SIM or STED microscope), which we do not have access to. Performing these experiments will require establishment of new external collaborations, which means an unpredictable amount of work and time. Besides, quantifying protein dynamics within sub-diffraction limit puncta would be a very challenging task if feasible at all. Thus, this work is more appropriate for a future study.

Nevertheless, although we are not immediately able to differentiate the fixation artifacts on potential small, confocal-invisible puncta and on large puncta, our confocal microscopy data sufficiently supported our conclusion that the apparent LLPS behavior of DsRed2-TAF15(IDR) is overall enhanced by PFA fixation.

Not all proteins analyzed showed a different distribution in living versus fixed cells (as shown in Figure 3 for GFP-FUS). The differences in the number and size of condensates observed in living and fixed cells should be correlated with the dynamic of the protein analyzed by FRAP. Are proteins with the highest mobility measured by FRAP corresponding to those that show an increased number/size of puncta upon fixation? Are FUS protein-protein interactions stable and less dynamic compared with the overall fixation rate?

We thank the reviewer for these important questions and agree that the severity of the fixation artifact should be correlated with a protein’s dynamics in the cell. Our kinetic model of fixation concerns protein-protein interaction dynamics instead of protein diffusion dynamics. Indeed, the model predicts that proteins with more dynamic interactions have more fixation artifacts, if the proteins of interest have similar absolute fixation rates. Since FRAP dynamics are contributed by both diffusion and interaction dynamics, which are often not separable or quantifiable from FRAP measurements (https://www.cell.com/biophysj/fulltext/S0006-3495(04)74392-1), we instead used SPT to directly measure protein interaction dynamics in the revised manuscript. As described in Response 2 and Response 17, we measured the dissociation rates of two proteins from respective puncta using SPT: Halo-TAF15(IDR), which is poorly preserved by fixation, and TAF15(IDR)-Halo-FTH1, which is well preserved by fixation. We found that the dissociation rate of Halo-TAF15(IDR) is much faster than that of TAF15(IDR)-Halo-FTH1, demonstrating more dynamic interactions of Halo-TAF15(IDR) than TAF15(IDR)-Halo-FTH1. Our observations that Halo-TAF15(IDR) has more dynamic interactions and is more poorly preserved by fixation than TAF15(IDR)-Halo-FTH1 agrees very well with our model’s prediction. Please see Response 2 and Response 17 for more details. Our new data and discussion have been added to the revised manuscript in Paragraph 3 on Page 13 and in Figure 3B, Figure 3E, Figure 6, and Video 2.

Recommendations for improving the writing and presentation: Defining whether a protein undergoes LLPS is based on different assays in vitro, using recombinant proteins and in cells. The observation that a given protein forms "puncta" inside the cells is generally not accepted as a criterion to establish whether it undergoes LLPS. The measure of the number, size, sphericity, liquid-like dynamic behavior (e.g. by FRAP), and sensitivity to agents such as e.g. hexanediol are all assays required to establish and characterize whether a given protein undergoes LLPS. These aspects should be described in the introduction of the paper. It is a bit simplistic to only focus on the calculation of the number and size of puncta before/after fixation.

We thank the reviewer for their suggestions. We agree that although detecting puncta in immunofluorescence images have been the most widely used approach for detecting LLPS for endogenous proteins, many other assays have been used for characterizing LLPS for overexpression systems in cells. We have now included the commonly used LLPS characteristics in cells in the Introduction (Paragraph 2 on Page 2) while describing our motivation of examining the effects of cell fixation on LLPS appearance: “Whereas rigorous characterization of LLPS in vivo has been challenging and remains a question under active investigation (McSwiggen et al., 2019b), detection of discrete puncta that have a spherical shape, undergo fusion and fission, and dynamically exchange biomolecules with the surrounding according to FRAP is often considered evidence of putative LLPS in living cells. While such diverse measurements have been widely used for studying proteins under overexpression conditions, far fewer approaches are available to probe LLPS under physiological conditions. Detecting local high-concentration regions or puncta of an endogenously expressed protein using immunofluorescence of fixed cells has been used in many studies as evidence of LLPS... Not only is the detection of puncta an inconclusive metric for establishing LLPS, whether a punctate distribution observed in fixed cells actually represents the live-cell scenario remains unclear…”